# DINCAE 1.0: a convolutional neural network with error estimates to reconstruct sea surface temperature satellite observations

Alexander Barth[1], Aida Alvera-Azcárate[1], Matjaz Licer[2], and Jean-Marie Beckers[1]

[1]GHER, University of Liège, Liège, Belgium
[2]National Institute of Biology, Marine Biology Station, Piran, Slovenia

**Correspondence:** A. Barth (a.barth@uliege.be)

**Abstract.**

A method to reconstruct missing data in sea surface temperature data using a neural network is presented. Satellite observations working in the optical and infrared bands are affected by clouds, which obscure part of the ocean underneath. In this paper, a neural network with the structure of a convolutional auto-encoder is developed to reconstruct the missing data based on the available cloud-free pixels in satellite images. Contrary to standard image reconstruction with neural networks, this application requires a method to handle missing data (or data with variable accuracy) in the training phase. The present work shows a consistent approach which uses the satellite data and its expected error variance as input and provides the reconstructed field along with its expected error variance as output. The neural network is trained by maximizing the likelihood of the observed value. The approach, called DINCAE (Data-Interpolating Convolutional Auto-Encoder) is applied to a 25-year time-series of Advanced Very High Resolution Radiometer (AVHRR) sea surface temperature data and compared to DINEOF (Data Interpolating Empirical Orthogonal Functions), a commonly used method to reconstruct missing data based on an EOF decomposition. The reconstruction error of both approaches is computed using cross-validation and in situ observations from the World Ocean Database. DINCAE results have lower error, while showing higher variability than the DINEOF reconstruction.

## 1 Introduction

The ocean temperature is an essential variable to study the dynamics of the ocean because density is a function of temperature and therefore the ocean velocity variability depends partially on ocean temperature. The amount of heat stored in the ocean is also critical for weather predictions at various scales (e.g. hurricane path prediction in the short range, as well as for seasonal and climate predictions).

The ocean sea surface temperature (SST) has been routinely measured since the beginning of the 1980s. However, as for any measuring technique working in the infrared or visible bands, clouds often obscure large parts of the field-of-view. Several techniques have been proposed for reconstructing gappy satellite data, but often small scale information is filtered out.

DINEOF (Data Interpolating Empirical Orthogonal Functions, Beckers and Rixen, 2003; Alvera-Azcárate et al., 2005), is an iterative method to reconstruct missing observations reducing noise in satellite datasets using empirical orthogonal functions (EOF). A truncated EOF decomposition using the leading EOFs is performed and the initially missing data are reconstructed using this EOF decomposition. The EOF decomposition and reconstruction is repeated until convergence. DINEOF has been applied to several oceanographic variables, at different spatial resolutions (e.g. Alvera-Azcárate et al. (2005) for SST, Alvera-Azcárate et al. (2007) for ocean colour, Alvera-Azcárate et al. (2016) for Sea Surface Salinity), providing accurate reconstructions. A truncated EOF decomposition will focus primarily in spatial structures with a "strong" signature (or more formally defined with a significant L2 norm compared to the total variance). Small scale structures can be included in a truncated EOF decomposition as long as their related variance is large enough to be present in the retained EOF modes. But small scale structures tend to be transient (short-lived) and therefore are often not retained in the dominant EOF modes. It should be noted that there is no explicit spatial filtering scale in DINEOF removing small scales (unlike other methods like optimal interpolation, kriging, spline interpolation). But in practice a similar smoothing effect is noticed because of the EOF truncation (which is necessary in the presence of clouds).

Neural networks are mathematical models that can efficiently extract nonlinear relationships from a mapping problem (i.e. an input/output relationship that can be determined through a mathematical function). Neural networks are therefore specially well positioned to learn nonlinear, stochastic features measured at the sea surface by satellite sensors, and their use might prove efficient in retaining these structures when analysing satellite data, for example for reconstructing missing data.

Neural networks can be composed of a wide variety of building blocks, such as fully connected layers (Rosenblatt, 1958; Widrow and Hoff, 1962) recurrent networks (e.g. Long Short-term Memory (Hochreiter and Schmidhuber, 1997), Gated recurrent units (Cho et al., 2014)) convolutional layers (LeCun et al., 1998; Krizhevsky et al., 2012). Recurrent networks work typically with a one dimensional list of inputs of a variable length (such as a text sentence). Fully connected layers and convolutional layers require to have a full dataset without missing data, at least for the training phase. For a review on neural networks the reader is referred to Schmidhuber (2015) and references therein. As neural networks are typically applied on a large and complete data set (i.e. no or almost no gaps) as input data, a solution needs to be found to handle a large number of missing data.

The use of neural networks in the frame of Earth Observation has been increasing recently. Garcia-Gorriz and Garcia-Sanchez (2007), for example, used meteorological variables like wind and air temperature (among others) to infer SST, with the aim of reproducing annual and interannual variability of SST during the pre-satellite era. Patil and Deo (2017) used a wavelet neural network to predict SST at various locations in the Indian Ocean, which allowed to focus on daily variations of SST. Pisoni et al. (2008) resorted to past instances and averaging to overcome gaps in SST, which results in smooth reconstructions. Krasnopolsky et al. (2016) used neural networks to infer ocean colour in the complete absence of these data (i.e. emulating a sensor failure). The neural network by Krasnopolsky et al. (2016) uses as input satellite sea surface elevation, sea surface salinity, sea surface temperature and *in situ* Argo salinity and temperature vertical profiles with some auxiliary infor-

mation (like longitude, latitude and time) to estimate the Chlorophyll-a concentration. The network does not use measured Chlorophyll-a concentration at a given location as input during inference (the reconstruction phase), nor the information from nearby grid points to infer Chlorophyll-a concentration. The network is exposed to the chlorophyll-a measurements only during the training phase. Jo et al. (2018) infers ocean colour from related data (SST and wind among others), taking advantage of

the close relation between different ocean variables, but also at a lower spatial resolution. Renosh et al. (2017) produced a suspended particulate matter dataset from model and in situ data using Self Organizing Maps, that was compared to satellite data. Chapman and Charantonis (2017) used surface satellite data to infer subsurface ocean currents also using Self Organizing Maps. Also using Self Organizing Maps, Jouini et al. (2013) reconstructed missing data in chlorophyll-a data using the relation between this variable and ocean currents (proxied by SST and sea surface height).

The objective of this manuscript is to present a neural network in the form of a convolutional auto-encoder which can be trained on gappy satellite observations, in order to reconstruct missing observations and also to provide an error estimate of the reconstruction. This neural network is referred to in the following as DINCAE (data-interpolating convolutional auto-encoder). An auto-encoder is a particular type of network which can compress and decompress the information in an input

dataset (Hinton and Salakhutdinov, 2006), effectively reducing the dimensionality in the input data. Projecting the input data on a low-dimensional subspace is also the central idea of DINEOF, where it is achieved by an EOF decomposition.

In section 2, the SST dataset used in this study is presented. This dataset is the input of the neural network described in section 3. This section includes the general structure of the network, the activation functions, skip connections, the cost function and

its optimization. The SST dataset is also reconstructed with DINEOF (section 4). The results are validated by cross-validation and by a comparison to the World Ocean Database 2018 in section 5. Finally, the conclusions are presented in section 6.

## 2 Data availability

For this study we used the longest available time series coming from the Advanced Very High Resolution Radiometer (AVHRR) dataset (Kilpatrick et al., 2001) spanning 25 years, from 1 April 1985 to 31 December 2009. The data are distributed by the

Physical Oceanography Distributed Active Archive Center (PODAAC), and have a spatial resolution of 4 km and a temporal resolution of 1 day. The dataset can directly be accessed by following the DOI link in the references (AVHRR Data). In this study, we focus on part of the Provençal basin (4.5625$^o$E, 9.5$^o$E and 39.5$^o$N, 44.4375$^o$N, Figure 1) where the main circulation features are the Western Corsican Current (WCC) and the Northern Current (NC) describing a cyclonic circulation pattern. In addition, several mesoscale and submesoscale circulation features are present in this area. With a resolution of 4 km, the SST

data measures only mesoscale and basin-wide variability.

For this study, only SST data with quality flags of 4 or higher are retained (Evans et al., 2009). One single image is composed of 112 x 112 grid points. If a given pixel has measurements less than 5% of the time, then it is not reconstructed and it is

considered as a land point in the following. In total, 27% of grid points correspond to land. Images with at least 20% of valid sea points are retained for the reconstruction which corresponds to a total of 5266 time instances.

To assess the accuracy of the reconstruction method, cross-validation is used (*e.g.* Wilks, 1995). For cross-validation a subset

of the data is withheld from the analysis and the final reconstruction is compared to the withheld dataset to access its accuracy. Since clouds have a spatial extent, we wanted to withhold data with a similar spatial structure. In the last 50 images we removed data according to the cloud mask of the first 50 images of the SST time series. The last 50 images represent the data from 2009-09-25 to 2009-12-27 (since some scenes with too few data have been dropped as mentioned before). These data are not used at all during either the training or the reconstruction phases, and can therefore be considered independent. In total,

106 816 measurements (i.e. individual pixels) have been withheld this way.

Initially, the average cloud coverage of the dataset is 46% (over all 25 years). The cloud coverage for the 50 last scenes is increased to 77% when the cross-validation points are excluded. A significant part of the scene is obscured after marking the data for cross-validation, but in the Mediteranan Sea the cloud coverage is relatively low compared to the globally average

cloud coverage which is 75% (Wylie et al., 2005). Removing some data for cross-validation makes the cloud coverage thus more similar to the global average.

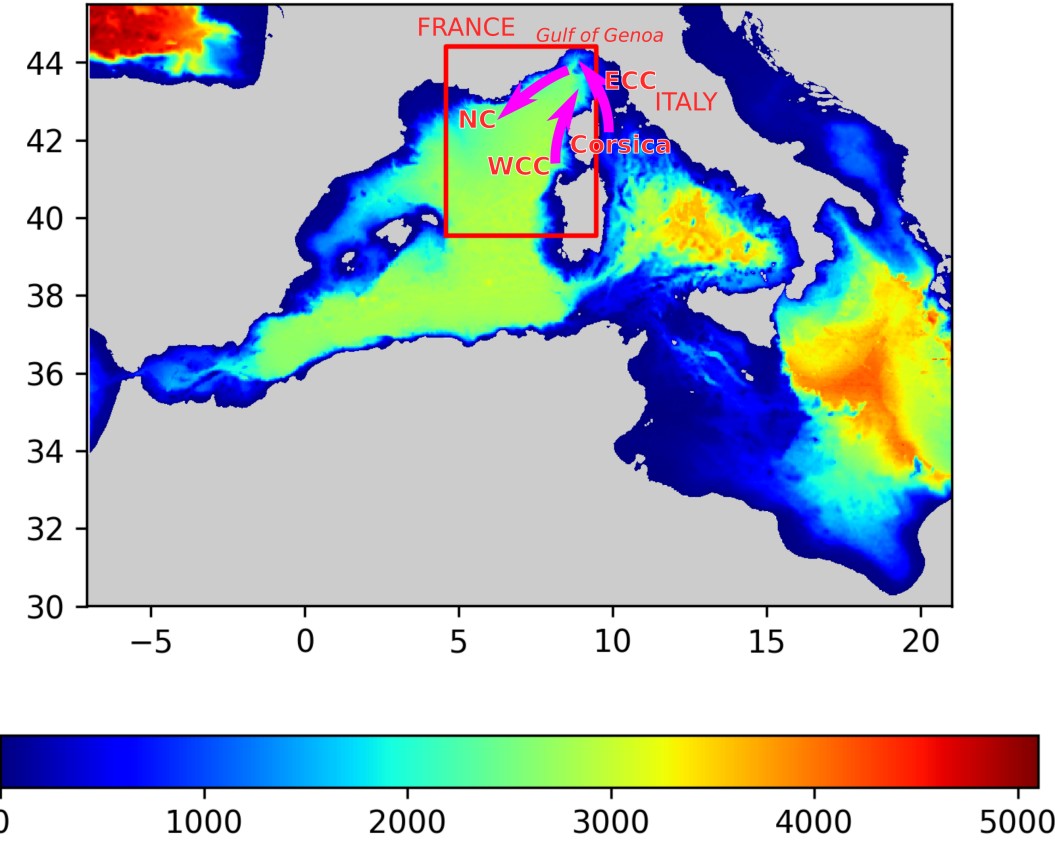

**Figure 1.** The red rectangle delimits the studied region and the color represents the bathymetry in meters. The arrows represent the main currents: the Western Corsican Current (WCC), the Eastern Corsican Current (ECC) and the Northern Current (NC)

## 3 Neural network with missing data as input

Convolutional and other deep neural networks are extensively used in computer vision and find an increasing number of applications in Earth sciences (Rasp et al., 2018; Bolton and Zanna, 2019; Zhou et al., 2016; Geng et al., 2015) where full datasets are available, at least for training a network. However, when using satellite data, the number of images without any clouds is very small and it is difficult to provide enough training data when only clear images are used. Therefore the aim is to derive a reconstruction strategy which can cope with the large amounts of missing data typically found in remote sensing data.

The handling of missing data is done in analogy to data assimilation in numerical ocean models. The standard optimal interpolation equations (*e.g.* Bretherton et al., 1976; Buongiorno Nardelli, 2012) can be written as follows:

$$\mathbf{P}^{a^{-1}}\mathbf{x}^a = \mathbf{P}^{f^{-1}}\mathbf{x}^f + \mathbf{H}^T\mathbf{R}^{-1}\mathbf{y}^o \tag{1}$$

$$\mathbf{P}^{a^{-1}} = \mathbf{P}^{f^{-1}} + \mathbf{H}^T\mathbf{R}^{-1}\mathbf{H} \tag{2}$$

where $\mathbf{x}^f$ is the model forecast with error covariance $\mathbf{P}^f$, $\mathbf{y}^o$ are the observations with error covariance $\mathbf{R}$ and $\mathbf{H}$ is the observation operator extracting the observed part from the state vector $\mathbf{x}^f$. The analysis $\mathbf{x}^a$ is the combined estimate with the error covariance matrix $\mathbf{P}^a$. We use these equations as an analogy to propose an approach to handle missing data (or data with errors varying in space and/or time). The main input datasets of the CAE are i) the SST divided by its error variance (corresponding to $\mathbf{R}^{-1}\mathbf{y}^o$) and ii) the inverse of the error variance (corresponding to the diagonal elements of $\mathbf{R}^{-1}$, assuming

spatially uncorrelated errors). If a data point is missing, then the corresponding error variance is considered infinitely large and the value at this point would be zero for both input datasets. The main difference is that in optimal interpolation, the observation vector $\mathbf{y}^o$ is multiplied by the inverse of the error covariance (possibly including non-diagonal elements) while in the present case we use only the error variance. The structure of the neural network will be used to spatially propagate the information from the observations.

    The time average has been removed from the SST dataset (computed over all years but excluding the cross-validation dataset). The neural network works thus with anomalies relative to this mean SST. To obtain reasonable results, the network uses more input than merely SST divided by its error variance and the inverse of the error variance. The total list of input parameters is consequently the following:

– SST anomalies scaled by the inverse of the error variance (the scaled anomaly is zero if the data is missing)

    – Inverse of the error variance (zero if the data is missing)

    – Scaled SST anomalies and inverse of error variance of the previous day

    – Scaled SST anomalies and inverse of error variance of the next day

    – Longitude (scaled linearly between -1 and 1)

– Latitude (scaled linearly between -1 and 1)

    – cosinus of the day of the year divided by 365.25

    – sinus of the day of the year divided by 365.25

The complete dataset is thus represented by an array of the size 10 x 112 x 112 x 5266 (number of inputs, number of grid points in the zonal direction, number of grid points in the meridional direction, number of time instances). The inverse of the error variance is either zero (for missing data) or a constant. The precise value of this constant is not important because it will be multiplied by a weight matrix and this weight matrix will be optimized by training the network. In future studies, it would be interesting to use sensor specific error statistics provided with GHRSST products, *i.e.* spatially and temporally varying error estimate. Using the previous and next day as inputs and the information on the season (last two inputs) will allow for a temporal coherency of the results. It should be noted that DINEOF does not use the day of the year of each satellite image, but it uses a temporal filter which increases the temporal coherence of the reconstruction (Alvera-Azcárate et al., 2009). The final layer of the neural network produces the following output:

– SST scaled by the inverse of the expected error variance

– Logarithm of the inverse of the expected error variance

The overall structure of the neural network (Table 2) is a convolutional autoencoder (CAE; Hinton and Salakhutdinov, 2006; Ronneberger et al., 2015). Its main building blocks are convolutional layers (LeCun et al., 1998; Krizhevsky et al., 2012). DINCAE uses 5 encoding and 5 decoding layers with a different number of filters. Beside the input and output layers, the number of filters are 16, 24, 36 and 54 (the number of filters increases 50% from one encoding convolutional layer to the next). All convolutional layers have a receptive field of 3x3 grid points (Simonyan and Zisserman, 2015). Between the convolutional layers there are max pooling or average pooling layers (Scherer et al., 2010) to progressively reduce the spatial resolution by only retaining either the maximum or average value of a region of 2x2 grid points. After the last encoding convolutional layer, there are two fully connected layers (the so-called bottleneck). The number of neurons in the bottleneck is a fifth of the number of the last pooling layer of the encoder (rounded to the nearest integer). Drop-out is used in the fully connected layers to avoid overfitting. The decoding layers are composed of convolutional layers and interpolation layers (to the nearest neighbor) to upsample the results. We also added skip connections between the output of pooling layers and the upsampling layers (Ronneberger et al., 2015). These skip connections correspond to layers 16, 19, 22 and 25 of Table 1. The motivation of this choice is that large scale information of the SST would be captured by the neurons in the bottle-neck, but small scale structures unrelated to the overall structure in the SST would be handled by these skip connections. In the absence of the skip connections, the small scale structures would be removed from the dataset.

A rectified linear unit (RELU) activation function is commonly used in neural networks which is defined as:

$$f(x) = \max(x, 0) \tag{3}$$

However, in our case it leads quickly (in 10 epochs) to a zero gradient and thus to no improvements in training. This problem is solved by choosing a leaky RELU (Maas et al., 2013) for the convolutions and the standard RELU for the fully connected layers.

$$f(x) = \max(x, \alpha x) \tag{4}$$

5     where we use here $\alpha = 0.2$. The output of the network, i.e. the 26th layer of Table 1, is an array $T_{ijk}^{(26)}$ with 112 x 112 x 2 elements. The first slice $k = 1$ is essentially interpreted as the logarithm of the inverse of the expected error variance and the second slice is the temperature anomaly divided by the error variance. The reconstructed temperature anomaly $\hat{y}_{ij}$ and the corresponding error variance $\hat{\sigma}_{ij}^2$ (for every single grid point $i, j$) are computed as:

$$\hat{\sigma}_{ij}^2 = \frac{1}{\max(\exp(\min(T_{ij1}^{(26)}, \gamma)), \delta)} \tag{5}$$

10    $$\hat{y}_{ij} = T_{ij2}^{(26)} \hat{\sigma}_{ij}^2 \tag{6}$$

where $\gamma = 10$ and $\delta = 10^{-3}\,^{\circ}\mathrm{C}^{-2}$. The $\min$ and $\max$ functions in the previous equations are introduced to avoid a division by a value close to zero or a floating point overflow. The effective range of the error standard deviation is thus from $\exp(-\gamma/2) = 0.0067\,^{\circ}\mathrm{C}$ to $\delta^{-\frac{1}{2}} = 31.6\,^{\circ}C$ which is a relatively wide range as the error is expected to be O(0.1) to O(1) $^o$C. The bounds are only effective during the very first epochs of the neural network where the weights are still close to random 15   values.

## 3.1   Training of the neural network

The input data set is randomly shuffled (over the time dimension) and partitioned into so-called mini-batches of 50 images, as an array of the size 10 x 112 x 112 x 50. The complete time series is splitted into 105 minibatches with 50 images each and one last minibatch with only 16 images (representing a total of 5266 as mentioned before). The splitting of the dataset 20   into minibatches is necessary because the graphical processing unit (GPU) has only a limited amount of memory. Computing the gradient over randomly chosen subsets also introduces some stochasticity which prevents the minimization algorithm from being trapped in a local minima. An optimization cycle using all 106 minibatches is called an epoch.

For every input image, more data points were masked (in addition to the cross-validation) by using a randomly chosen cloud 25   mask during training. The cloud mask of a training image would thus be the union of the cloud mask of the input dataset and a randomly chosen cloud mask. This allows us to assess the capability of the network to recover missing data under clouds. Without the additional clouds, the neural network would simply learn to reproduce the SST values that are already received as

**Table 1.** List of all steps in DINCAE. The additional dimension of the size of the minibatch is omitted in the output sizes below. Max pooling and average pooling are tested for the pooling layers.

| number | type | output size | parameters |
|---|---|---|---|
| 1 | input | 112 x 112 x 8 | |
| 2 | conv. 2d | 112 x 112 x 16 | n. filters = 16, kernel size = (3,3) |
| 3 | pooling 2d | 56 x 56 x 16 | pool size = (2,2), strides = (2,2) |
| 4 | conv. 2d | 56 x 56 x 24 | n. filters = 24, kernel size = (3,3) |
| 5 | pooling 2d | 28 x 28 x 24 | pool size = (2,2), strides = (2,2) |
| 7 | conv. 2d | 28 x 28 x 36 | n. filters = 36, kernel size = (3,3) |
| 8 | pooling 2d | 14 x 14 x 36 | pool size = (2,2), strides = (2,2) |
| 9 | conv. 2d | 14 x 14 x 54 | n. filters = 54, kernel size = (3,3) |
| 10 | pooling 2d | 7 x 7 x 54 | pool size = (2,2), strides = (2,2) |
| 11 | fully connected layer | 529 | |
| 12 | drop-out layer | 529 | drop-out rate for training = 0.3 |
| 13 | fully connected layer | 2646 | |
| 14 | drop-out layer | 2646 | drop-out rate for training = 0.3 |
| 15 | nearest neighbor interpolation | 14 x 14 x 54 | |
| 16 | concatenate output of 15 and 8 | 14 x 14 x 90 | |
| 17 | conv. 2d | 14 x 14 x 36 | n. filters = 36, kernel size = (3,3) |
| 18 | nearest neighbor interpolation | 28 x 28 x 36 | |
| 19 | concatenate output of 18 and 5 | 28 x 28 x 60 | |
| 20 | conv. 2d | 28 x 28 x 24 | n. filters = 24, kernel size = (3,3) |
| 21 | nearest neighbor interpolation | 56 x 56 x 24 | |
| 22 | concatenate output of 21 and 3 | 56 x 56 x 40 | |
| 23 | conv. 2d | 56 x 56 x 16 | n. filters = 16, kernel size = (3,3) |
| 24 | nearest neighbor interpolation | 112 x 112 x 16 | |
| 25 | concatenate output of 24 and 1 | 112 x 112 x 26 | |
| 26 | conv. 2d | 112 x 112 x 2 | n. filters = 2, kernel size = (3,3) |

input. At every epoch a different mask is applied to a given image to mitigate overfitting and aid generalization.

The aim of DINCAE is to provide a good SST reconstruction but also an assessment of the accuracy of the reconstruction. The output of the neural network is assumed to be a Gaussian probability distribution function (pdf) characterized by a mean $\hat{y}_{ij}$ and a standard deviation $\hat{\sigma}_{ij}$. Given this pdf one can compute the likelihood $p(y_{ij}|\hat{y}_{ij}, \hat{\sigma}_{ij})$ of the observed values $y_{ij}$. The weights and biases in the neural network are adjusted to maximize the likelihood of all observations. Maximizing the likelihood is equivalent to minimizing the negative log-likelihood:

$$J(\hat{y}_{ij}, \hat{\sigma}_{ij}) = -\frac{1}{N} \sum_{ij} \log\left(p(y_{ij}|\hat{y}_{ij}, \hat{\sigma}_{ij})\right) \tag{7}$$

where $N$ is the number of measurements in $y_{ij}$ (excluding therefore land points and cross-validation points). Including the number measurements $N$ is important as it can change from one mini-batch to the other. The likelihood of the observations $p(y_{ij}|\hat{y}_{ij}, \hat{\sigma}_{ij})$ is given by a Gaussian distribution:

$$p(y_{ij}|\hat{y}_{ij}, \hat{\sigma}_{ij}) = \frac{1}{\sqrt{2\pi\hat{\sigma}_{ij}^2}} \exp\left(-\frac{(y_{ij} - \hat{y}_{ij})^2}{2\hat{\sigma}_{ij}^2}\right) \tag{8}$$

5      The cost function has finally the following form:

$$J(\hat{y}_{ij}, \hat{\sigma}_{ij}) = \frac{1}{2N} \sum_{ij} \left[ \left(\frac{y_{ij} - \hat{y}_{ij}}{\hat{\sigma}_{ij}}\right)^2 + \log(\hat{\sigma}_{ij}^2) + 2\log(\sqrt{2\pi}) \right] \tag{9}$$

The loss function per individual scalar sample is the term in brackets of the previous equation. The first term is directly related to the mean square error, but scaled by the estimated error standard deviation. The second term penalizes any over-estimation of the error standard deviation. The third term is a constant term which can be neglected in the following as it does 10   not influence the gradient. The sum in the previous equation runs over all grid points where a measurement is available but excluding the measurements withheld for cross-validation as the later are never used during training.

We used the Adam optimizer (Kingma and Ba, 2014) with the standard parameters for the learning rate $\alpha = 0.001$, the exponential decay rate for the first moment estimates $\beta_1 = 0.9$, and for the second-moment estimates $\beta_1 = 0.999$, regularization 15   parameter $\epsilon = 10^{-8}$.

During the development of the neural network, it was clear that it tended to overfit the provided observations, leading to degraded results when comparing the network to cross-validation data. Commonly used strategies were therefore used to avoid overfitting, namely introducing a drop-out layer between the fully connected layers of the network. The drop-out layer ran-20   domly sets, with a probability of 0.3, the output of these intermediate layers to zero during the training of the network. We also added some Gaussian-distributed noise to the input of the network with a zero mean and a standard deviation of $0.05^oC$.

It is useful to compare the proposed approach to the traditional autoencoder to highlight the different choices that have been adopted. The essential steps to implement and validate an auto-encoder are the following:

25    – Some data are marked for validation and never used during training

     – The network is given some data as input and produces an output which should be as close as possible to the input. All training data are given thus at all epochs to the network

     – The network is validated using the validation data set aside.

In essence, the traditional auto-enoder optimises how well the provided input data can be recovered after dimensionality reduction. In the present approach, there are two steps where data are intentionally hidden to the network:

1. The validation data that were set aside and never used during the training, similar to the traditional auto-encoder.

2. Some additional data in every minibatch were set aside to compute the reconstruction error and its gradient (unlike the traditional auto-encoder). This additional subset is chosen at random.

This is done because the main purpose of the network is to assess the ability of the network to reconstruct the missing data using the available data. The proposed method is not withholding less data than the traditional auto-encoder. The downside of the approach is that the cost function fluctuates more because it is computed only over a relatively smaller set of data. But for us this is acceptable (and controlled by taking the average of the output of the network at several epochs, as explained later) because the cost function reflects more closely the objective: reconstructing missing data from the available data (instead of reproducing the input data as it is the case of the traditional auto-encoder).

The traditional auto-encoder approach trained using only clear images was not considered because only 13 images of out 5266 have a cloud coverage of less than 5%. So the ability to handle missing data was a requirement for us from the start.

## 4 DINEOF reconstruction

The results of the DINCAE method are compared to the reconstruction obtained by the DINEOF method (Alvera-Azcárate et al., 2005) which uses an EOF-basis to infer the missing data. As a first step, the spatial and temporal mean is removed from the data, and the missing data are set to zero. The leading EOF modes are then computed and the missing data are reconstructed using these EOFs (Alvera-Azcárate et al., 2005). A temporal low-pass filter with a cut-off period of 1.08 days is applied to improve the temporal coherency of the results, following Alvera-Azcárate et al. (2009). This filter effectively propagates the information in time so that for a given date the satellite data from the previous and next days are used in the reconstruction. The optimal number of EOFs retained in this reconstruction is 13 modes, which explain 99.4% of the variability of the initial data.

The classical DINEOF technique reconstructs the cross-validation data points withheld in the last 50 images with an error of 0.4629$^o$C and a slight negative bias of -0.0922 $^o$C (Table 2). As only 13 modes are retrained by DINEOF for the reconstruction, some small scale structures are smoothed-out, which is a well known property of a truncated EOF decomposition (Wilks, 1995). This smoothing effect results in an RMS (root mean square) error of 0.3864$^o$C when comparing the reconstructed dataset to all the initially present SST (*i.e.* used for the reconstruction). A somewhat surprising result is that when using less data with DINEOF (only from the last two years, *i.e.* 2008 to 2009), 19 EOFs modes are retained, leading to a reconstruction with richer structures. Therefore, the RMS error compared to all the initially present SST provided to DINEOF (but excluding the

**Table 2.** Comparison with the independent cross-validation data and the dependent data used for training (in $^o$C). CRMS is the centered root mean square error.

| | CV data | | | non-CV data | | |
| --- | --- | --- | --- | --- | --- | --- |
| | RMS | CRMS | bias | RMS | CRMS | bias |
| DINEOF | 0.4629 | 0.4536 | -0.0922 | 0.3864 | 0.3864 | -0.0029 |
| DINEOF (2008-2009) | 0.4789 | 0.4715 | -0.0839 | 0.3376 | 0.3375 | -0.0038 |
| DINCAE (no skip connections) | 0.4458 | 0.4456 | 0.0147 | 0.2957 | 0.2953 | 0.0153 |
| DINCAE (2 skip connections) | 0.4222 | 0.4217 | 0.0198 | 0.1519 | 0.1504 | -0.0210 |
| DINCAE (all skip connections) | 0.3900 | 0.3895 | 0.0199 | 0.1383 | 0.1380 | -0.0097 |
| DINCAE (all skip connections - median) | 0.3922 | 0.3918 | 0.0190 | 0.1342 | 0.1342 | -0.0012 |
| DINCAE (all skip connections and wider layers) | 0.4005 | 0.4003 | 0.0147 | 0.1339 | 0.1328 | 0.0175 |
| DINCAE (all skip connections and narrower layers) | 0.3928 | 0.3915 | 0.0318 | 0.1379 | 0.1345 | -0.0300 |
| DINCAE (all skip connections and 5 conv. layers) | 0.4603 | 0.4557 | -0.0648 | 0.1396 | 0.1364 | -0.0295 |
| DINCAE (all skip connections and 3 conv. layers) | 0.3991 | 0.3990 | 0.0083 | 0.1350 | 0.1346 | -0.0101 |
| DINCAE (all skip connections and average pooling) | 0.3835 | 0.3834 | 0.0102 | 0.1251 | 0.1250 | -0.0063 |

cross-validation data) is lower ($0.3375^o$C) than when the 25 years dataset is used. However, the RMS error compared to the cross-validation data is slightly worse with the 2 years dataset ($0.4789^o$C). As the main validation statistic for this study is the RMS error compared to the cross-validation dataset, we use the DINEOF reconstruction of the full 25 years dataset. DINEOF used 15 hours of a single core on an Intel Core i7-3930K CPU to reconstruct the 25 years.

5 ## 5 Results

The figure 2 shows the cost function for every minibatch. Large fluctuations are quite apparent from this figure. But it is expected that the cost function will fluctuate using any optimization method based on mini-batch (unless the learning rate explicitly is decreased to zero, which is not the case here) because the cost function is evaluated using a different mini-batch at every iteration. Consequently, the gradient of the cost function also includes some stochastic variability. Even if the dataset

10 is small and the gradient could be computed over the entire dataset at once, using mini-batches is still advised because these fluctuations allow the cost function to get out of a local minima (Ge et al., 2015; Masters and Luschi, 2018). While the mini-batch selection effectively computes the gradient over a temporal subset, the additional data marked as missing within a minibatch is a spatial subset which enhances these fluctuations but allows us to define the cost function more closely to our objective (i.e. inferring the missing data from observations, as explained above).

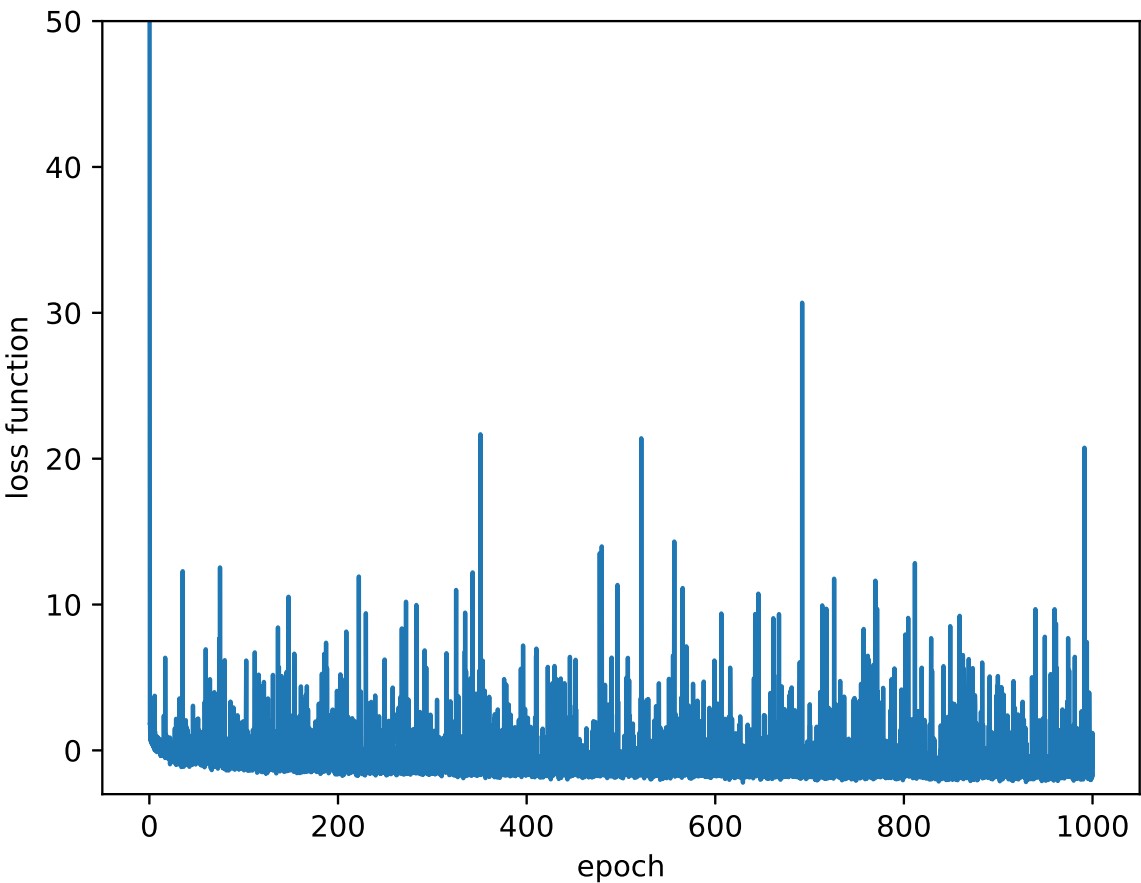

**Figure 2.** The cost function computed internally for every minibatch during the optimization.

The neural network is updated using the gradient for every mini-batch during training and after every 10 epochs the current state of the neural network is used to infer the missing data over the whole time series, and in particular reconstructing the missing data is the cross-validation dataset. But importantly, the network is not updated using the cross-validation data.

5    Figure 3 shows the RMS error relative to the cross-validation dataset computed every ten epochs (during this reconstruction phase drop-out is disabled) using DINCAE. There is an initial sharp decrease of the cross-validation error and after 200 epochs the RMS error has mostly stabilized but still presents some fluctuations. These fluctuations are due to the fact that the gradient computed at every optimization set is computed over a subset of the data and this subset varies at every optimization step. As mentioned before, in every minibatch a random subset (in form of clouds) of data is marked as missing and the gradient
10    is computed over this randomly changing subset which leads to some fluctuations in the gradient and thus in the parameters

of the neural network. In order to obtain a better estimate of the reconstruction, we average the output of the neural network between epoch 200 and epoch 1000 (saved at every 10th epoch) which leads to a better reconstruction than every individual intermediate results. The expected error of the reconstruction is similarly averaged. Ideally, one would take the correlation of the error between the different reconstructions into account. Ignoring these error correlations could result in overestimating the expected error of the reconstruction. Alternatively one would average the output of an ensemble of neural networks initialized with different weights (and possibly using different structures) but this would significantly increase the necessary computing resources of the technique (Krizhevsky et al., 2012). But this ensemble averaging approach could be beneficial to improve the representation of the expected error and the accuracy of the reconstruction.

Instead of using the average, the median reconstruction was also tested, as the median is more robust to outliers. The results were very similar and slightly better with the average instead of the median SST. In the following, only the average estimate is used.

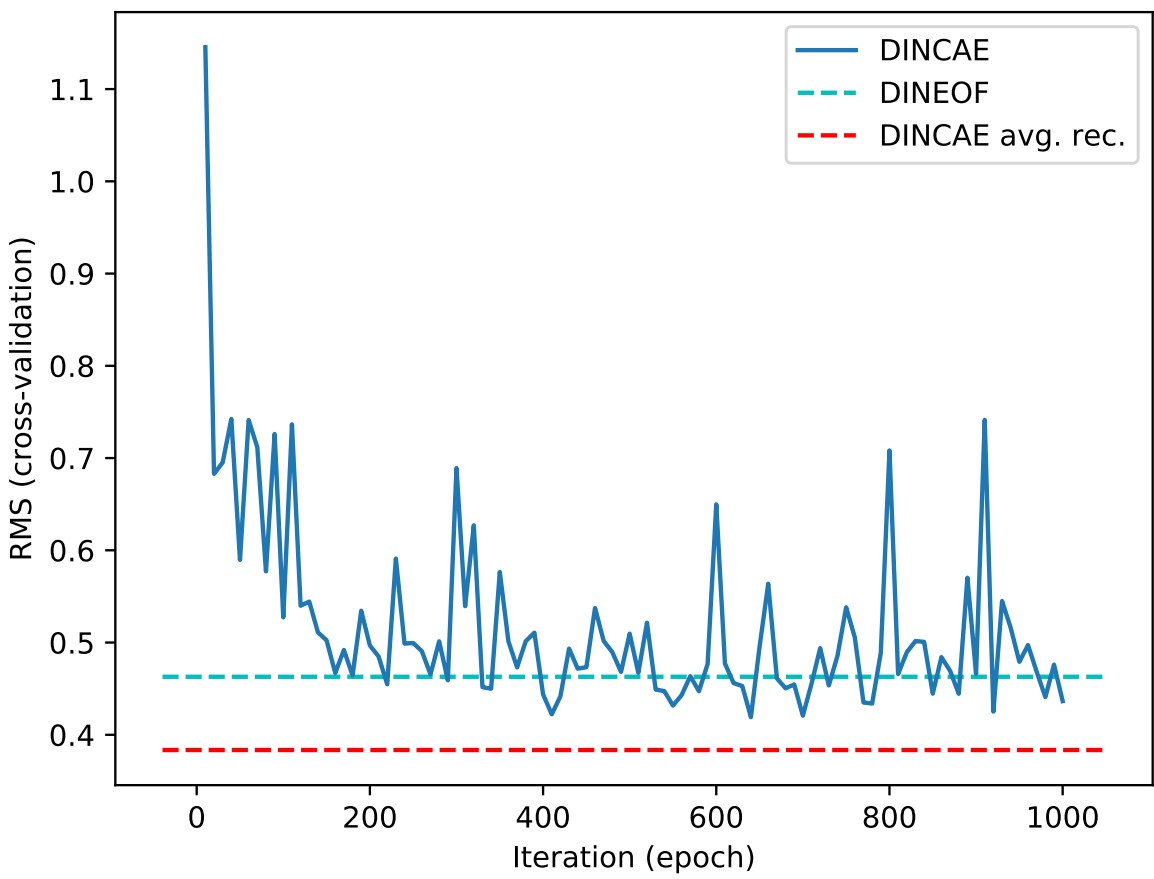

**Figure 3.** RMS difference with cross-validation dataset as a function of iteration. The solid blue line represents the DINCAE reconstruction at different steps of the iterative minimization algorithm. The dashed cyan line is the DINEOF reconstruction and the dashed red line is the average DINCAE reconstruction between epoch 200 and 1000.

Training this network for 1000 epochs takes 4.5 hours on a GeForce GTX 1080 and Intel Core i7-7700 with the neural network library tensorflow (Abadi et al., 2015). For a trained network, reconstructing all 25-year takes only 8 seconds. All computations are done in single precision.

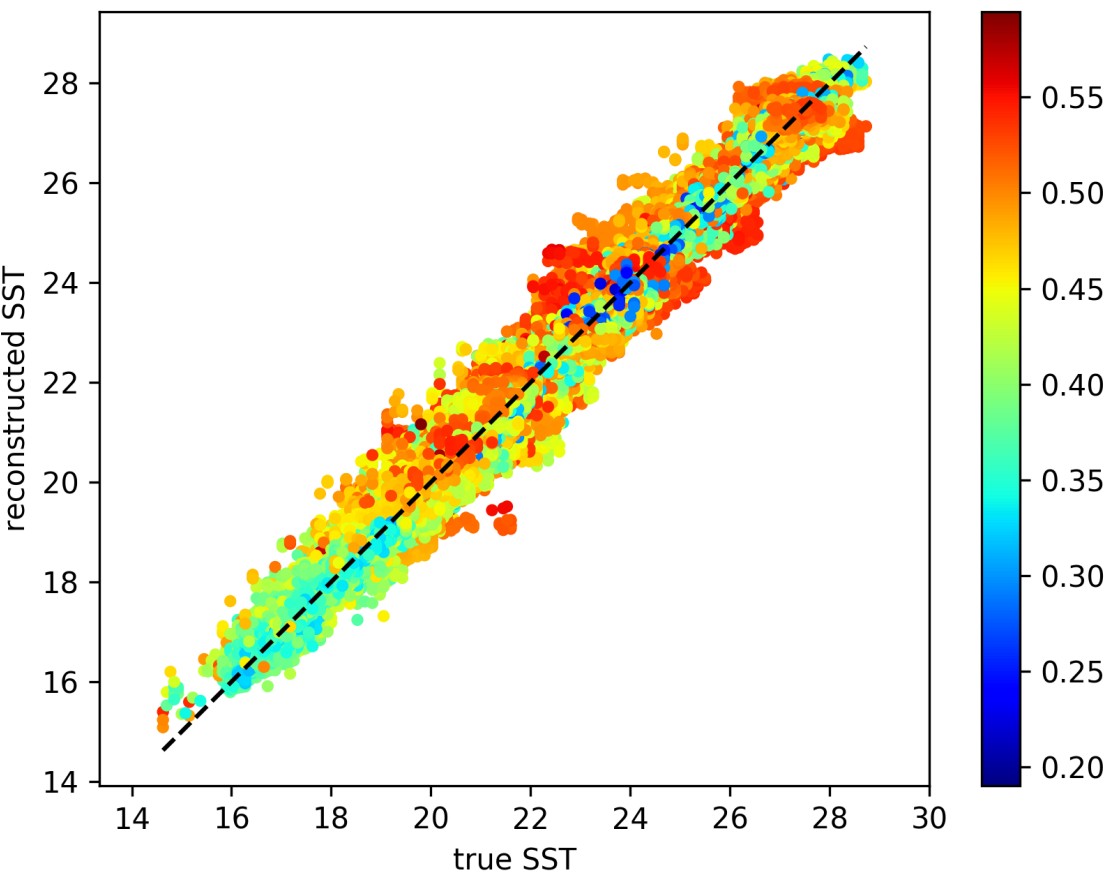

**Figure 4.** The original SST versus the reconstructed SST for the cross-validation dataset. The color represents the estimated expected error standard deviation.

Figure 4 shows a scatter plot of the true SST (withheld during cross-validation) and the corresponding reconstructed SST. The color represents the estimated expected error standard deviation of the reconstruction. Low error values are expected to be closer to the dashed line. Reconstructed and cross-validation SST tends to cluster relatively well around the ideal dashed line. Typically the lower expected errors are found more often near the dashed line than at the edge of the cluster of points.

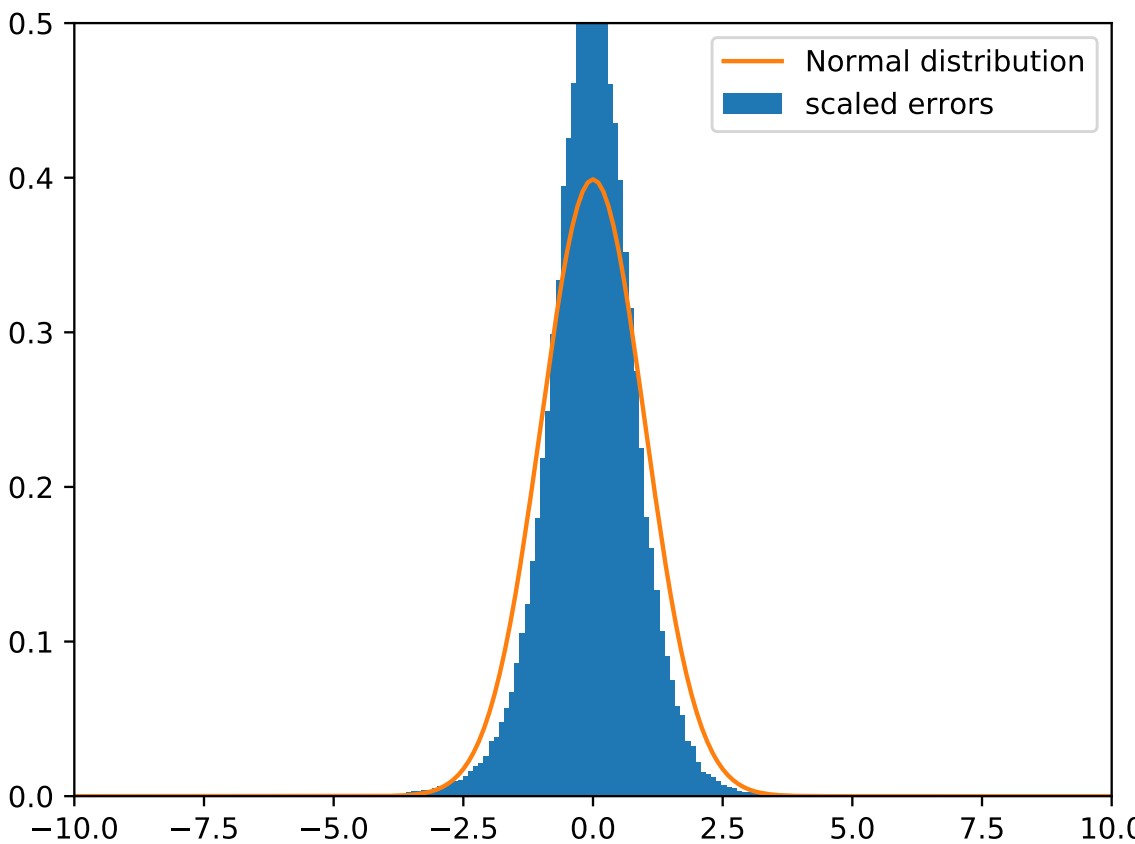

**Figure 5.** Scaled errors are computed as the difference between the reconstructed SST and the actual measured SST (withheld during cross-validation) divided by the expected standard deviation error.

To obtain a clearer idea of the reliability of the expected error we computed the difference between the cross-validation SST and the reconstructed SST divided by the expected error standard deviation. A histogram of the scaled differences is shown in Figure 5. The scaled error follows the theoretical distribution relatively well. When a Gaussian pdf is fitted to the histogram of the scaled error, one obtains a mean of -0.02 and a standard deviation 0.85 (both adimensional), so that generally speaking DINCAE is overestimating the expected error by 15 %.

An interpolation technique which is commonly used in operational context, is optimal interpolation. This technique is able to provide an expected error variance of the interpolated fields based on a series of assumptions, in particular that the errors are Gaussian distributed with a known covariance and zero mean. Given these assumptions, the error variance of the optimal interpolation algorithm is only found to be weakly related to the observed RMSE in a study of Pisano et al. (2016) using

satellite sea surface temperature in the Mediterranean Sea. The averaged results of DINCAE overestimate the actual error by 15% which in this context can be seen as an improvement.

Different variants of the neural network are tested in order to optimize its structure. The number of skip connections has a quite significant impact on the results. The cross-validation RMS error is reduced from 0.4458 $^o$C (no skip connections), to 0.4222 $^o$C with 2 skips connections (layer 22 and 25 of Table 1), and further to 0.3900 $^o$C with all skip connections between the encoder and decoder layer of the same size. At the same time, the RMS error relative to the used data (i.e. data not reserved for cross-validation) measuring the degree of smoothing is reduced from 0.2957 $^o$C (no skip connections) to 0.1383 $^o$C with all skip connections.

Increasing the number of filters of the convolutional layers from 16, 24, 36, 54 to 16, 32, 48, 64 (with the input convolution layer fixed by 10 filters as it has to correspond to the number of inputs) and increasing the number of neurons of the bottleneck accordingly leads to a slight degradation for the present case compared to the cross-validation dataset, which indicates that the neural network starts to overfit if the number of filters is increased. A subsequent test with narrower convolutional layers of size 16, 22, 31 and 44 lead to very similar but slightly worse results with 0.3928 $^o$C.

The DINCAE neural network with an increasing or decreasing number of layers (5 or 3 convolutional layers) did not improve the results. However, it is possible that the depth of the neural network is dependent on the available training data set and that for a more extensive data, increasing the number of layers could have a positive effect.

Max pooling layers are commonly used in image classification problems (*e.g.* Simonyan and Zisserman, 2015; Krizhevsky et al., 2012) where the strongest detected feature is passed from one layer to the next. However, the purpose of this network here is rather different as we intend to recover missing data which requires to spread the information spatially. Therefore we also tried the network with average pooling instead of max pooling, which further reduced the reconstruction error to 0.3835 $^o$C. This better performance of average pooling can be related to the fact that SST images do generally not have as abrupt gradients as typical images used for classification. Another way to look at this is the fact that for a dynamical system in the linear regime, different flow features (solutions to the underlying primitive equations) coexist and contribute in an additive way to the total flow.

For every time instance we use the data from 3 time instances in the reconstruction: the current day, as well the data from the previous and next day. As a variant of the previous reconstruction experiment we increase the number of time instances from 3 to 5 centered at the current time instance. However, the cross-validation error for this experiment is 0.433 $^o$C and the results are not improved. Increasing the number of input features can aggravate the potential for overfitting as the number of parameters in the neural network is increased. Here the number of parameters is increased by 40% in the first convolutional layer. A combination of convolutional neural network with recurrent neural networks (like Long Short-Term Memory, LSTM)

might be a better way to include the time dependencies.

In all cases the biases are relatively small and the present discussion is essentially also valid when considering the centered RMS (i.e. the RMS difference when the bias is removed). In the following, we only use DINCAE with all skip connections and 4 convolutional layers with a number of filters of 16, 24, 36, 54 and average pooling for future comparison.

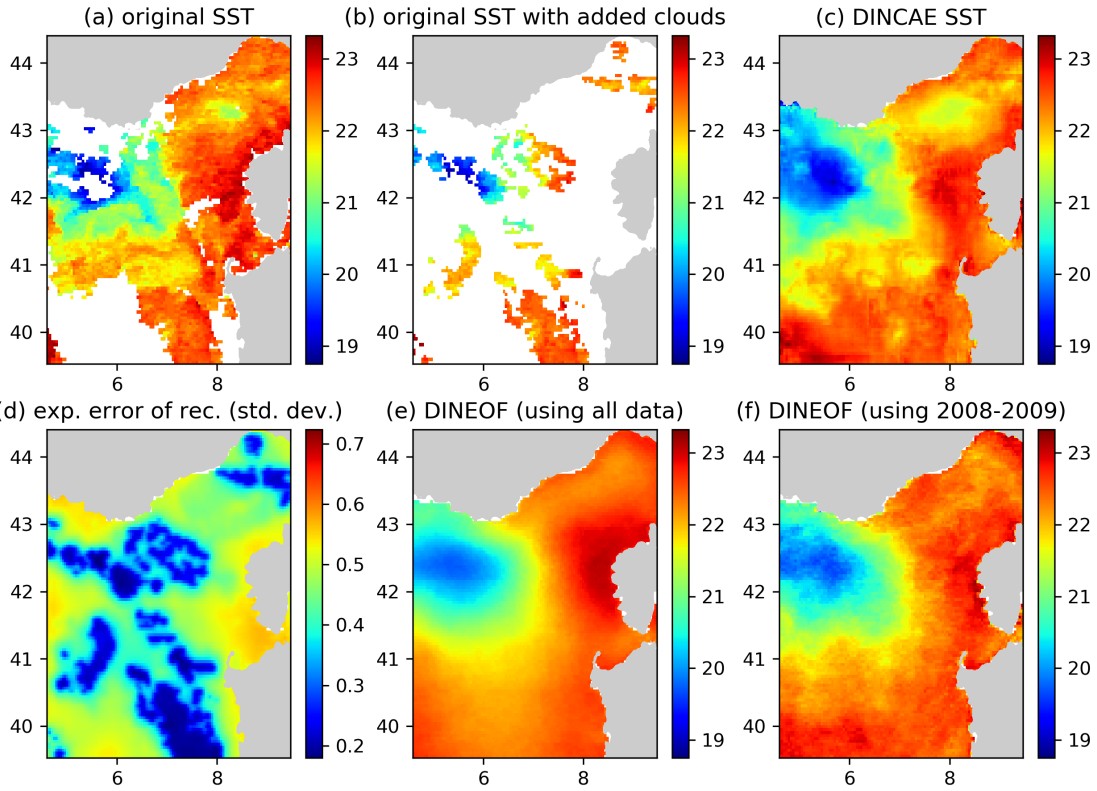

**Figure 6.** Panel (a) the original AVHRR SST, (b) AVHRR SST with additional clouds for cross-validation, (c) the DINCAE reconstruction, (d) the expected error variance of the DINCAE reconstruction, (e) the DINEOF reconstruction using all data, (f) the DINEOF reconstruction using only the data from 2008-2009. All panels are in degrees Celsius and valid for the date 2009-10-13.

Figure 6 shows the SST reconstructions for 13 October 2009. The overall SST structure is reasonable in all reconstructions. The cold water in the western part of the domain is better defined in the DINCAE reconstruction, and the general position of the 21$^o$C isotherm agrees better with the SST observations in the DINCAE reconstruction than with the DINEOF results.

**Table 3.** Comparison with the World Ocean Database for SST grid points covered by clouds. The RMS, CRMS and bias are in degree Celsius.

|        | RMS    | CRMS   | bias    |
|--------|--------|--------|---------|
| DINEOF | 1.1676 | 1.1102 | -0.3616 |
| DINCAE | 1.1362 | 1.0879 | -0.3278 |

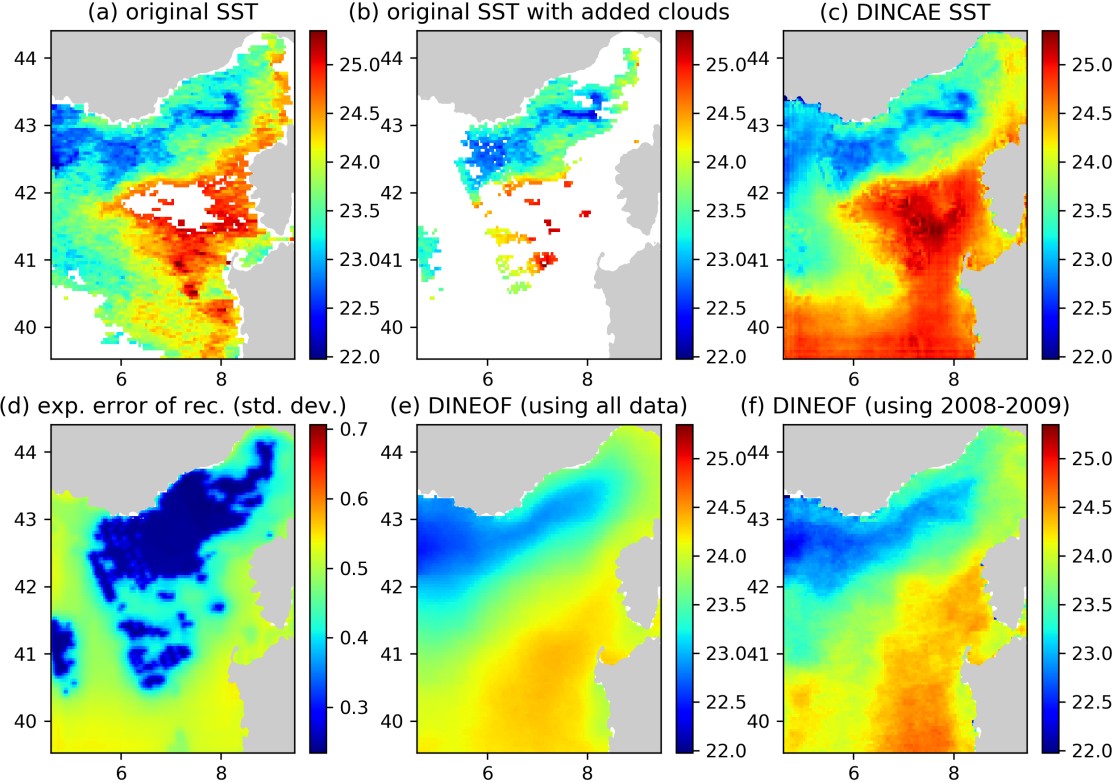

**Figure 7.** Panel (a) the original AVHRR SST, (b) AVHRR SST with additional clouds for cross-validation, (c) the DINCAE reconstruction, (d) the expected error variance of the DINCAE reconstruction, (e) the DINEOF reconstruction using all data, (f) the DINEOF reconstruction using only the data from 2008-2009. All panels are in degrees Celsius and valid for the date 2009-09-29.

In some cases the DINCAE reconstruction also introduces some artefacts as some zonal and meridional gradients near the open boundaries (Figure 7). This is probably due to the fact that in the convolutional layers, zero padding is applied so that the convolution operation does not change the size of the arrays. As this issue is relatively localized at the border it is recommended that one chooses a slightly larger domain than the primary domain of interest for the reconstruction.

To further quantify how well the reconstruction methods could recover data under a cloud cover, we use in situ temperature from the World Ocean Database 2018 (Boyer et al., 2018). For every in situ grid point, the SST image with the same time stamp (ignoring hour, minutes and seconds) is interpolated to the nearest grid cell relative to the location of the in situ observations. Only in situ observations corresponding to a cloudy SST pixel are used in the following. In total, there are 774 surface in situ observations. The depth of the in situ observations should be between 0.5 m and 1 m and if there are multiple data points between this depth range, the data point closest to the surface is used. As expected, biases play a more important role now when comparing in situ observations with reconstructed satellite data (Table 3). DINCAE represented a small improvement relative to the DINEOF reconstruction confirming the results from the cross-validation comparison.

In Figure 8 the variability of the reconstructed SST dataset is assessed. These figures represent the standard deviation relative to a yearly average climatology computed for the original SST, and the reconstructions from DINCAE and DINEOF. For the original SST, the climatological mean SST and the standard deviation were computed only using the available data. The standard deviation derived from DINCAE matches well the standard deviation from the original data, in particular in the interior of the domain, but the standard deviation is too large along the southern coast of France and Corsica. The DINEOF standard deviation matches the original SST standard deviation better in those areas but generally underestimates the SST standard deviation. Given the fact that DINCAE tends to retain more variability in the reconstruction it is thus remarkable that it still features a lower RMS despite the so-called double penalty effect (Gilleland et al., 2009; Ebert et al., 2013), *i.e.* RMS-based error measures tend to be lower for smoother fields with lower variability, but this is not the case here.

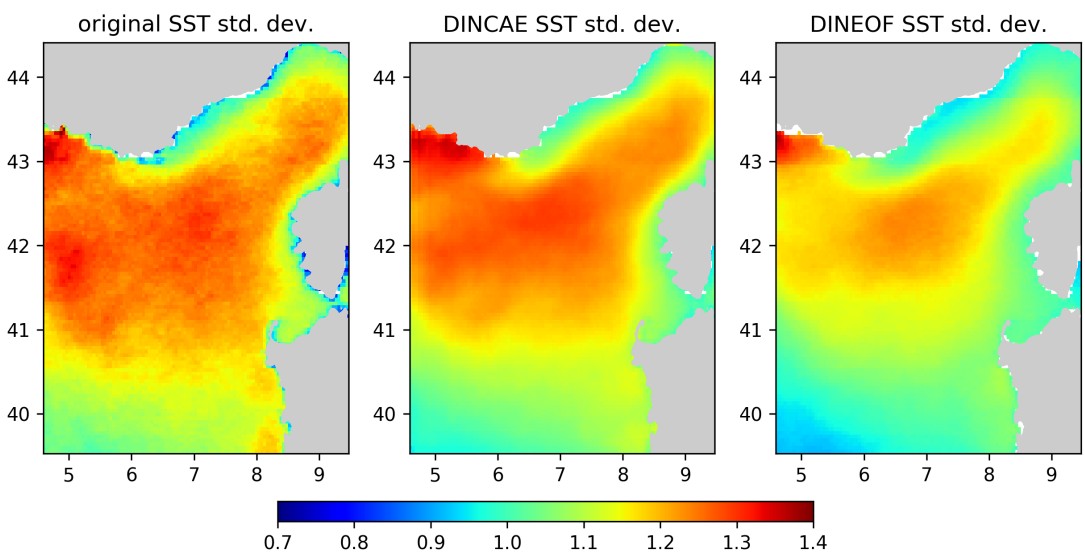

**Figure 8.** Standard deviation computed around the seasonal average in degrees Celsius.

## 6    Conclusions

This paper presents a consistent way to handle missing data in satellite images for neural networks. Essentially, the neural network uses the measured data divided by its expected error variance. Missing data are thus treated as data with an infinitely large error variance. The cost function of the neural network is chosen such that the network provides the reconstruction but also the

confidence of the reconstruction error (quantified by the expected error variance). An over- or underestimation of the expected error variance are both penalized by maximising the likelihood and assuming Gaussian distributed errors. This approach can be easily generalized to parametric probability distributions, in particular to log-normal distributions for concentrations like remote sensed chlorophyll-a concentration or suspended sediment concentration.

The presented reconstruction method DINCAE compared favourably to the widely used DINEOF reconstruction method which is based on a truncated EOF analysis. Formally there are similarities between an auto-encoder (composed of just 2 fully-connected layers) and an EOF projection followed by an EOF reconstruction (Chicco et al., 2014). However, neural networks can represent non-linear relationships which is not possible with an EOF approach. Both methods were compared by

cross-validation and the DINCAE method resulted in RMS error reduction from 0.46$^o$C to 0.38$^o$C.

The expected error for the reconstruction reflects well the areas covered by the satellite measurements as well as the areas with more intrinsic variability (like meanders of the Northern Current). The expected error predicted by the neural network provides a good indication of the accuracy of the reconstruction.

The accuracy of the reconstructed data under clouds was also assessed by comparing the results to in situ observations of the World Ocean Database 2018. Also compared to this dataset, the RMS error of the DINCAE reconstruction is lower than the corresponding results from DINEOF.

It is quite common that data analysis methods to reconstruct missing data tend to smooth the available observations in order to fill the area of missing observations. Therefore, the temporal variability (relative to the seasonal cycle) of the reconstructed sea surface temperature was computed from the original data and from the reconstructed data using DINCAE and DINEOF. The variability of the reconstructed SST with DINEOF generally underestimated the variability in the original dataset, but the variability of the DINCAE reconstruction matched the variability of the original data relatively well.

The tests conducted in this paper show that DINCAE is able to provide a good reconstruction of missing data in satellite SST observations and retaining more variability than the DINEOF method. In addition, the expected error variance of the reconstruction is estimated avoiding several assumptions (difficult to justify in practice) of other methods like optimal interpolation.

*Code availability.* The source code is released as open source under the terms of the GNU General Public Licence v3 (or, at your option, any later version) and available at the address https://github.com/gher-ulg/DINCAE and https://doi.org/10.5281/zenodo.3251813.

*Author contributions.* A.B. designed and implemented the neural network. A.A.A made the DINEOF simulations. A.B., A.A.A., M.L. and J.M.B. contributed to the planning and discussions and to the writing of the manuscript.

*Competing interests.* The authors have no competing interests

*Acknowledgements.* We thank the anonymous reviewer #1, #2 and Zhaohui Han for reading carefully the manuscript, providing constructive remarks and interesting interpretations of the results.

The F.R.S.-FNRS (Fonds de la Recherche Scientifique de Belgique) is acknowledged for funding the position of Alexander Barth. This research was partly performed with funding from the Belgian Science Policy Office (BELSPO) STEREO III programme in the framework of

the MULTI-SYNC project (contract SR/00/359). Matjaz Licer would like to acknowledge COST action ES1402 - "Evaluation of Ocean Syntheses" for funding his contribution to this work. Computational resources have been provided in part by the Consortium des Équipements de Calcul Intensif (CÉCI), funded by the F.R.S.-FNRS under Grant No. 2.5020.11 and by the Walloon Region. The AVHRR v5 dataset was obtained from the NASA EOSDIS Physical Oceanography Distributed Active Archive Center (PO.DAAC) at the Jet Propulsion Laboratory, Pasadena, CA. The National Centers for Environmental Information (NOAA, USA) and the (International Oceanographic Data and Information Exchange (IODE) are thanked for the World Ocean Database 2018.

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
