# Peer review of "DINCAE 1.0: a convolutional neural network with error estimates to reconstruct sea surface temperature satellite observations"

_Geoscientific Model Development, 2019_

## Referee Comment (RC1) · Anonymous Referee #1 · 3 Dec 2019

DINCAE 1.0: a convolutional neural network with error estimates to reconstruct sea surface temperature satellite observations

General Comments: It is an interesting paper overall. The author uses Auto Encoder to reconstruct the missing data commonly found in optical satellite remote sensing caused by instrument failure or cloud cover. The author uses an interesting way to handle missing data in training image. As compared to the widely used DINEOF method, the author showed that DINCAE can, on some degree, produce better results measured in RMSE metrics, as well as spatial distributions of SST. From the technique side, Auto Encoder is a commonly used machine learning method in semantic segmentation and

object detection. The author uses this method to solve, particularly SST, reconstruction obtained from satellite remote sensing. It is an interesting and meaningful problem to tackle. But as for developing a new methodology, I have some concerns.

(1)The applicability of this method to other satellite measurements. Variables such as SST, have low frequency variability both in space and time (If I am right). This nature suggest that it is relatively easier for CNN to estimate the spatial correlation (e.g. for an image in which there are multiple people, it is harder to do segmentation than for an image with only lawn and sky ). This also gives ground that average pooling turns out to achieve better results than max pooling, as stated in paper. For variables, especially those on land, such as plant reflectance, usually have high frequency variability both in space and time due to heterogeneous growth stage, backgroud, and so on. These feature creates additional challenges, which I think, cannot be handled with the method configuration stated in paper. It will be interesting to see how it does (This may not directly related to the topic of this paper). Additionally, the method is tested at one site, which hardly persuasive to show its applicability over the globe. Will a model trained at one site be able to use at another site, or it is needed to develop a new model to a new site, which usually needs a lot of work to prepare data, training model, parameter tuning and so on? If so, from model deployment side, what the advantage of using it?

(2)Temporal feature of reconstructed variables EOF method is essentially PCA analysis. DINEOF method does take into consideration of both temporal and spatial correlation of variables, to my understanding. Though DINCAE, as described in the paper, also uses the spatial and temporal correlation of variables, it only uses correlation presented in 3 days (the day, the day before and the day after). In other words, spatial information is what it uses mainly for reconstructing. Do you have persuasive arguments that 3 days correlation in time are enough to capture temporal dependency? However, longer time dependency, e.g. seasonality, may also be important on estimating missing values. In this case, net work configuration both capture spatial and temporal structure of variables (e.g. LSTM + CNN) could be more general and powerful.

Technique Comments: Page 1 line 2: 'A method to reconstruct missing data in satellite data using a neural network is presented' The first sentence is not as precise as it should be. As the first impression that this paper is going to introduce a neural network based method to reconstruct/interpolate gappy satellite images caused by cloud coverage, instrument failures (e.g. LandSat 7) and so on. However the following paper mostly discussed an AutoEncoder method to reconstruct SST and tested only on SST.

Page 2 line 31: 'effectively reducing....' What is the meaning of putting this sentence here

Page 4. Figure 1 caption 'The arrow represent...' There is no arrow on figure

Page 4 line 6: 'so that for a given date also the satellite' Delete 'also'

Page 5 line 12, '...in the following' Delete 'in the following'

Page 5 line 19 'assimilation of data' Change to 'data assimilation'

Page 7 line 20 'skip connection' Does the resolution of SST data have effect on how you use skip connections? How large scale is called large scale for resolution of 4KM by 4KM, how about SST with resolution 1KM by 1KM. From another point of view, this operation again consolidate to use the spatial information for reconstruction, while temporal information somehow is ignored.

Page 8 line 5 The two parameters here seemly have profound effect on reconstruction result, how does these two parameter chosen?

Page 9 line 15 'the output of the neural network is a Gaussian probability distribution' The author assume the output is a Gaussian distribution, 'is a Gaussian distribution' means the author know it is Gaussian.

Page 10 line 18-21 'As mentioned before, ....neural network' Not quite understand the training procedure here. 'a random subset of data is marked as missing'? Since the missing data is marked randomly for each epoch, it is possible that at epoch = k, some

part of data is marked as missing, while at epoch = k+1, the same part of data of the same image is marked as available. If this is the case, it essentially means the model was told what it should predict randomly? This is somewhat contradictory with Page 9 line 10.

Page 10 line 21-22 'we average ...intermediate result' Why do not average multiple runs?

Page 11 Figure 2 caption 'red dash line ...' How come the average DINCAE reconstruction is smaller than RMSE at any given epoch? Also, the error curve indicates that the model has no sign of convergence. I bet if you continue training the model for another 1000 epochs, the cross validation error curves will not converge. This also indicates that there might be something wrong in the training procedure. Can you plot your loss function here as well?

Page 14 line 16 'also tried ...' The max pooling operation tries to extract distinct signals from neighbors, while average pooling operation tries to extract common signals from neighbors. For SST, which has low frequency variation in space, it makes sense average pooling should do better than max pooling.

Page 17 line 14 '...reconstruction is it thus...' Change 'is it' to 'it is'
* * *

---

## Author Comment (AC1) · 11 Dec 2019

**We would like to thank the reviewer for carefully reading the manuscript and providing constructive criticism. We hope that our reply below answers the questions and comments by the reviewer in an adequate manner. Our response is in bold-face.**

General Comments: It is an interesting paper overall. The author uses Auto Encoder to reconstruct the missing data commonly found in optical satellite remote sensing caused by instrument failure or cloud cover. The author uses an interesting way to handle

missing data in training image. As compared to the widely used DINEOF method, the author showed that DINCAE can, on some degree, produce better results measured in RMSE metrics, as well as spatial distributions of SST. From the technique side, Auto Encoder is a commonly used machine learning method in semantic segmentation and object detection. The author uses this method to solve, particularly SST, reconstruction obtained from satellite remote sensing. It is an interesting and meaningful problem to tackle. But as for developing a new methodology, I have some concerns.

1. The applicability of this method to other satellite measurements. Variables such as SST, have low frequency variability both in space and time (If I am right). This nature suggest that it is relatively easier for CNN to estimate the spatial correlation (e.g. for an image in which there are multiple people, it is harder to do segmentation than for an image with only lawn and sky). This also gives ground that average pooling turns out to achieve better results than max pooling, as stated in paper. For variables, especially those on land, such as plant reflectance, usually have high frequency variability both in space and time due to heterogeneous growth stage, background, and so on. These feature creates additional challenges, which I think, cannot be handled with the method configuration stated in paper. It will be interesting to see how it does (This may not directly related to the topic of this paper). Additionally, the method is tested at one site,which hardly persuasive to show its applicability over the globe. Will a model trained at one site be able to use at another site, or it is needed to develop a new model to a new site, which usually needs a lot of work to prepare data, training model, parameter tuning and so on? If so, from model deployment side, what the advantage of using it?

   **We think that it is quite normal to first apply a technique to a limited area before addressing the problem at global scale and we choose a parameter which is quite important for oceanography. The initial papers of DINEOF**

(Beckers and Rixen, 2003; Alvera-Azcárate et al., 2005) focused also on sea surface temperature reconstruction and in subsequent papers it was shown that it can also be applied to other ocean parameters with (some adaptations) such as Chlorophyll concentration, sea surface salinity and sea surface currents. We are envisioning a similar path for the DINCAE technique. As our background is in physical oceanography, we can only speculate whether this technique can be applied to land-based data. The underlying motivation of DINEOF (and to some extent of DINCAE) is the fact that a large fraction ( 90% and more) of the variability of the ocean (as obtained from remote sensing data) can be explained by a reduced number modes (often 10 to 50 modes). So even when the satellite scene is partially obscured the missing data can be recovered using the data present in the satellite scene because the number of inherent degrees of freedom is relatively low. The reviewer mentions the case of plant reflectance which seems indeed to be a case where the number of degrees of freedom is apparently much higher and where it can be indeed difficult to use the same method. But we would say that this is a difficult case for any reconstruction method because the data actually measured by the satellite has less information on the data obscured by e.g. clouds. For such cases, it is particularly important to associate a reconstructed scene with a reliable error estimate. At least for sea surface temperature we were able to demonstrate that this could be done.

In the same way that the EOFs obtained from DINEOF are specific and only optimal to the studied zone, we see that the network of DINCAE is, until the contrary is proven, only specific to a given zone. The aim of this paper is to use the typical use-case of DINEOF (limited zone, ocean parameter) and to see if DINCAE can provide a better reconstruction than DINEOF. Many other reconstruction techniques used in oceanography, such as optimal

**interpolation or variational analysis, require some set of parameters to be tuned to a specific site. For DINCAE it is rather the structure of the network which can be optimized for a given site but there is arguably a greater chance that these depend less of the studied site than for example parameters like the correlation length in optimal interpolation. We actually have good results using the present network structure on the Adriatic Sea and we have been contacted by a researcher using the same network architecture on the South China Sea and West Philippine Sea providing a convincing reconstruction.**

**For this paper we worked on a regional scale, because we believe that this matches the typical approach of oceanographic studies which focus on a specific zone of interest and improve the understanding of the processes in this area (instead of trying to understand a process directly on a global scale).**

2. Temporal feature of reconstructed variables EOF method is essentially PCA analysis. DINEOF method does take into consideration of both temporal and spatial correlation of variables, to my understanding. Though DINCAE, as described in the paper, also uses the spatial and temporal correlation of variables, it only uses correlation presented in 3 days (the day, the day before and the day after). In other words, spatial information is what it uses mainly for reconstructing. Do you have persuasive arguments that 3 days correlation in time are enough to capture temporal dependency? However, longer time dependency, e.g. seasonality, may also be important on estimating missing values. In this case, network configuration both capture spatial and temporal structure of variables (e.g. LSTM + CNN) could be more general and powerful.

**The cloud cover varies normally quite rapidly from one day to the next as**

it does so on the time scale of a couple of hours as revealed by geostationary satellites like SEVIRI. For polar orbiting satellite we typically have a sea-surface image every day. So the one day before and one day after, is justified by the fact that we will have a reasonable chance that a pixel covered at a given day is not covered by the day before or the day after. Providing more than just 3 days could improve the performance as it would increase the available information, but it could also increase the risk of overfitting. As a response to the reviewer, we also tried with 5 time instances but it degraded the results. The following has been added to the manuscript:

For every time instance we use the data from 3 time instances in the reconstruction: the current day, as well the data from the previous and next day. As a variant of the previous reconstruction experiment we increase the number of time instance from 3 to 5 centered at the current time instance. However, the cross-validation error for this experiment is 0.433 °C and the results are not improved. Increasing the number of input features can aggravate the potential for overfitting as the number of parameters in the neural network is increased. A combination of convolutional neural network with recurrent neural networks (like Long short-term memory, LSTM) might be a better way to include the time dependencies.

The idea using an LSTM is indeed an interesting idea. But we rather think this should be addressed in a follow-up study as we were able to show progress using the present structure of the neural network.

The present technique uses also the day-of-the-year as input of the neural network so the information about the season is available to the neural network. The day-of-the-year is transformed by a cosinus and sinus specifically to facilitate the representation of the seasonal cycle (e.g. the 1st January is as close to the 2nd January as the 31st December).

**Technique Comments**

Page 1 line 2: 'A method to reconstruct missing data in satellite data using a neural network is presented' The first sentence is not as precise as it should be. As the first impression that this paper is going to introduce a neural network based method to reconstruct/interpolate gappy satellite images caused by cloud coverage, instrument failures (e.g. LandSat 7) and so on. However the following paper mostly discussed an AutoEncoder method to reconstruct SST and tested only on SST.

**While we think that the method is generic, we have only tested it on SST and thus we changed the abstract accordingly as suggested by the reviewer to make it clear that so far we only demonstrated its use for SST. The abstract now starts with:**

**A method to reconstruct missing data in sea surface temperature data using a neural network is presented.**

**This matches in fact the scope as set by the title of the manuscript which also mentioned specifically sea surface temperature.**

Page 2 line 31: 'effectively reducing....' What is the meaning of putting this sentence here.

**We think that the dimensionality reduction is a central aspect for the reconstruction of missing data. This aspect is actually shared with DINEOF. To make this clear we have expanded this paragraph.**

**An auto-encoder is a particular type of network which can compress and decompress the information in an input dataset (Hinton and Salakhutdinov, 2006), effectively reducing the dimensionality in the input data. Projecting the input data on a low-dimensional subspace is also the central idea of DINEOF, where it is achieved by an EOF decomposition.**

Page 4. Figure 1 caption 'The arrow represent...' There is no arrow on figure

**Unfortunately, we included an earlier version of the figure in the manuscript (without the arrows). The correct figure is the one below and the manuscript is updated.**

Page 4 line 6: 'so that for a given date also the satellite' Delete 'also'

**Ok, done, thanks.**

Page 5 line 12, '...in the following' Delete 'in the following'

**Ok, done, thanks.**

Page 5 line 19 'assimilation of data' Change to 'data assimilation'

**Ok, done, thanks.**

Page 7 line 20 'skip connection' Does the resolution of SST data have effect on how you use skip connections? How large scale is called large scale for resolution of 4KM by 4KM, how about SST with resolution 1KM by 1KM. From another point of view,this

operation again consolidate to use the spatial information for reconstruction, while temporal information somehow is ignored.

**For us, large-scale refers to scale of variability which affects the SST over the entire domain: for example the overall position of the main current and the heating and cooling related to the seasonal cycle. Short scale refers to mesoscale circulation features (visible also in SST) related to meanders and eddies which typically have a length-scale of 50 km.**

**The initial idea is that these large scales should go through the bottle-neck of the convolutional autoencoder while the small scales are handled by the skip connections (experiment labeled "DINCAE (2 skip connections)" in table 1). However, it turned out that it is beneficial to have these skip connections at all levels of the convolutional neural network (experiment labeled "DINCAE (all skip connections)") so that a distinction between scale with and without skip connections (and large versus small scale) is no longer necessary.**

Page 8 line 5 The two parameters here seemly have profound effect on reconstruction result, how does these two parameter chosen?

**It is not clear to us, why the reviewer thinks that these parameters have a profound effect on the reconstruction. The range of allowed values are very far from restrictive. We added the following to clarify this point.**

**The effective range of the error standard deviation is thus from $exp(-\gamma/2) = 0.0067$ °C to $\delta^{-\frac{1}{2}} = 31.6$ °C which is a relatively wide range as the error is expected to be O(0.1) to O(1) °C . The bounds are only effective during the very first epochs of the neural network where the weights are still close to random values.**

Page 9 line 15 'the output of the neural network is a Gaussian probability distribution'
The author assume the output is a Gaussian distribution, 'is a Gaussian distribution'
means the author know it is Gaussian.

**We agree and changed "is a Gaussian probability distribution" by "is assumed
to be a Gaussian probability distribution".**

Page 10 line 18-21 'As mentioned before, ....neural network' Not quite understand the
training procedure here. 'a random subset of data is marked as missing'? Since the
missing data is marked randomly for each epoch, it is possible that at epoch = k, some
part of data is marked as missing, while at epoch = k+1, the same part of data of the
same image is marked as available. If this is the case, it essentially means the model
was told what it should predict randomly? This is somewhat contradictory with Page 9
line 10.

**We agree that this part is confusing and more information is added to the
manuscript. First, we want to explain how a traditional auto-encoder works:**

- **Some data are marked for validation and never used during training**

- **The network is given some data as input and produce an output which
  should be as close as possible to the input. So all training data are given
  at all epochs to the network**

- **The network is validated using the validation data set aside.**

**So the traditional auto-enoder optimises how well the provided input data can be
recovered after dimensionality reduction.**

**In our approach, there are two steps where data are intentionally hidden to the
network:**

[Figure]

1. **The validation data that were set aside and never used during the training (page 3, line 19 of the original manuscript), similar to the traditional auto-encoder.**

2. **Some additional data in every minibatch were set aside to compute the reconstruction error and its gradient (unlike the traditional auto-encoder). This additional subset is chosen at random.**

**This is done because the main purpose of the network is to assess the ability of the network to reconstruct the missing data using the available data. In fact, we are not withholding less data than the traditional auto-encoder. The downside of the approach is that the cost function fluctuates more because it is computed only over a relatively smaller set of data. But for us this is acceptable (and controlled by taking the average of the output of the network at several epochs) because the cost function reflects more closely the objective: reconstruct missing data from the available data (instead of reproducing the input data as it is the case of the traditional auto-encoder).**

**The traditional auto-encoder approach trained using only clear images was not considered because only 13 images of out 5266 have a cloud coverage of less than 5%. So the ability the handle missing data was for us a requirement from the start.**

**Concerning the specific question "Since the missing data is marked randomly for each epoch, it is possible that at epoch = k, some part of data is marked as missing, while at epoch = k+1, the same part of data of the same image is marked as available. . . .". The reviewer is right in its interpretation. But this is always the case in supervised learning. The gradients are computed using observations (or true labels,...) of the training dataset and observations are used multiple times**

**(once in every epoch). But of course, the validation dataset is never used during training and used only at the last step to assess the accuracy of the network.**

Page 10 line 21-22 'we average ...intermediate result' Why do not average multiple runs?

**We agree that averaging of multiple runs would be preferable but it would increase tremendously the computation time, by e.g. a factor of 30 if one would average over 30 runs for example. We added the following to the manuscript:**

**Alternatively one would average the output of an ensemble of neural networks initialized with different weights (and possibly using different structures) but this would significantly increase the necessary computing resources of the technique (Krizhevsky et al., 2012). But this ensemble averaging approach could be beneficial to improve the representation of the expected error and the accuracy of the reconstruction.**

Page 11 Figure 2 caption 'red dash line ...' How come the average DINCAE reconstruction is smaller than RMSE at any given epoch? Also, the error curve indicates that the model has no sign of convergence. I bet if you continue training the model for another 1000 epochs, the cross validation error curves will not converge. This also indicates that there might be something wrong in the training procedure. Can you plot your lossfunction here as well?

**It is quite common that the RMS error relative to a cross-validation data of a neural network does not converge. This is actually the basis of strategies like early stopping Prechelt2012. The RMSE of the average DINCAE reconstruction is smaller than the RMSE at any given epoch because computing the RMSE is a non-linear operation. The DINCAE reconstruction at a given epoch included**

**some variability which is not (or insufficiently) constrained by the observations. This explains also why the CV RMSE fluctuates. By taking the mean of the reconstruction at different epoch these fluctuations are averaged out and a better reconstruction is obtained. An alternative technique would be the use of an ensemble of neural networks (Krizhevsky et al., 2012) as noted also by the reviewer in his/her other comment.**

**The figure below shows the loss function for every minibatch. High fluctuations are quite apparent from this figure. But it is expected that the loss function us­ing any optimization method based on mini-batch fluctuates (unless the learning explicitly is forced to zero, which is not the case here) because the loss function is evaluated using a different mini-batch at every iteration. Consequently the gradient of the cost function includes also some stochastic variability. Even if the dataset is small and the gradient could be computed over the entire dataset at once, using mini-batches is still advised because these fluctuations allow the cost function to get out of a local minima (Ge et al., 2015; Masters and Luschi, 2018). While the mini-batch selection effectively computes the gradient over a temporal subset, the additional data marked as missing within a minibatch is a spatial subset which enhances these fluctuations but allows us to define the cost function more closely to our objective (i.e. inferring the missing data from ob­servations, as explained above). (The previous paragraph has also been added to the manuscript).**

Page 14 line 16 'also tried ...' The max pooling operation tries to extract distinct sig­nals from neighbors, while average pooling operation tries to extract common signals from neighbors. For SST, which has low frequency variation in space, it makes sense average pooling should do better than max pooling.

**Thanks for this remark. We added this interpretation to the manuscript. In fact,**

in the current research literature max pooling has completely replaced average pool posed in the pioneering work from LeCun et al. (1998) for CNN and image recognition. It was a surprise to see that the seemingly outdated average pooling worked better than the max pooling for our case. But we agree with the interpretation of the reviewer which has been included in the revised manuscript. Another way to look at this is the fact that for a dynamical system in the linear regime, different flow features (solution to the underlying primitive equations) coexist and contribute in an additive way to the total flow.

Page 17 line 14 '...reconstruction is it thus...' Change 'is it' to 'it is

**Sorry for the typo, and thank you for reading the manuscript so carefully to the end!**

**References**

Alvera-Azcárate, A., Barth, A., Rixen, M., and Beckers, J.-M.: Reconstruction of incomplete oceanographic data sets using Empirical Orthogonal Functions. Application to the Adriatic Sea Surface Temperature., Ocean Modelling, 9, 325–346, https://doi.org/10.1016/j.ocemod.2004.08.001, http://hdl.handle.net/2268/4296, 2005.

Beckers, J.-M. and Rixen, M.: EOF calculation and data filling from incomplete oceanographic datasets, Journal of Atmospheric and Oceanic Technology, 20, 1839–1856, https://doi.org/10.1175/1520-0426(2003)020<1839:ECADFF>2.0.CO;2, 2003.

Ge, R., Huang, F., Jin, C., and Yuan, Y.: Escaping From Saddle Points - Online Stochastic Gradient for Tensor Decomposition, CoRR, abs/1503.02101, http://arxiv.org/abs/1503.02101, 2015.

Hinton, G. E. and Salakhutdinov, R. R.: Reducing the Dimensionality of Data with Neural Networks, Science, 313, 504–507, https://doi.org/10.1126/science.1127647, https://science.sciencemag.org/content/313/5786/504, 2006.

Krizhevsky, A., Sutskever, I., and Hinton, G. E.: ImageNet Classification with Deep Convolutional Neural Networks, in: Advances in Neural Information Processing Systems 25, edited by Pereira, F., Burges, C. J. C., Bottou, L., and Weinberger, K. Q., pp. 1097–1105, Curran Associates, Inc., http://papers.nips.cc/paper/4824-imagenet-classification-with-deep-convolutional-neural-networks.pdf, 2012.

Masters, D. and Luschi, C.: Revisiting Small Batch Training for Deep Neural Networks, CoRR, abs/1804.07612, http://arxiv.org/abs/1804.07612, 2018.

Prechelt, L.: Early Stopping — But When?, pp. 53–67, Springer Berlin Heidelberg, Berlin, Heidelberg, https://doi.org/10.1007/978-3-642-35289-8_5, 2012.

———————————————————

[Figure]

**Fig. 1.** The red rectangle delimits the studied region and the color represents the bathymetry in meters. The arrows represent the main currents: the Western Corsican Current (WCC), the Eastern Corsican Curren

**Fig. 2.** The loss function computed internally for every minibatch during the optimization.

[Figure]

---

## Referee Comment (RC2) · Anonymous Referee #1 · 13 Dec 2019

The author has clarified a lot. Thanks.

1."We actually have good results using the present network structure on the Adriatic Sea and we have been contacted by a researcher using the same network architecture on the South China Sea and West Philippine Sea providing a convincing reconstruction."

**This is indeed interesting to see**

2. "However, the cross-validation error for this experiment is 0.433 C and the results are not improved. Increasing the number of input features can aggravate the potential for overfitting as the number of parameters in the neural network is increased."

[Figure]

**The channels of input data increased from 8 to 10. But the filter size, the number of output feature maps, and layer size, number of layers stay the same. Thus, the parameters of the network should stay the same. Right?**

3. "The effective range of the error standard deviation is thus from..."

**This is helpful. Previously, the author introduced two variables with no explaination of what and why. Previously, from the formula only, it seems like the value of these two variables will affect strongly the computation. e.g. delta = 100 vs. delta = 0.01**

4."The RMSE of the average DINCAE reconstruction is smaller than the RMSE at any given epoch because computing the RMSE is a non-linear operation. The DINCAE reconstruction at a given epoch included..."

**I am not very sure I fully understand it. But I will leave it to other reviewers!**

5. "the additional data marked as missing within a minibatch is a spatial subset which enhances these fluctuations but allows us to define the cost function more closely to our objective..."

**I would guess the fundamental reason why the RMSE and Loss fluctuate so much is that the random mark missing data in every mini batch. Because in every epoch, the spatial correlation of missing and available data is disrupted due to random marking, hence what the network has learned in previous epoch is disrupted as well, which eventually is reflected in RMSE and Loss. The fluctuations may not have so much to do with mini-batch optimization. Perhaps one way to check is to use same random mark missing data for every 20 epochs, and average at every 20 epochs. Just my opinion.**

---

## Author Comment (AC2) · 17 Dec 2019

We thank the review again for this quick response and very insightful comments.

1. "This is indeed interesting to see"

**Thank you for your encouragement.**

2. The channels of input data increased from 8 to 10. But the filter size, the number

of output feature maps, and layer size, number of layers stay the same. Thus,the parameters of the network should stay the same. Right?

**In addition to the SST (divided by the expected error variance), we are also providing the inverse of the expected error variance. The number of input channels changed from 4+3\*2 = 10 to 4+5\*2 = 14. The filter size of the first convolution network stayed at 16 filters. While previously the 3x3 convolution was realized with 3 x 3 x 10 x 16 (width x height x input channels x output channels), in the version with more time instances the convolution matrix had the dimensions 3 x 3 x 14 x 16 (so a 40In the first submitted version of the manuscript we indeed wrote that the total size of the array is 8 x 112 x 112 x 5266. This should be 10 x 112 x 112 x 5266 and this is corrected in the revised version. We apologize for this confusion.**

3. This is helpful. Previously, the author introduced two variables with no explaination of what and why. Previously, from the formula only, it seems like the value of these two variables will affect strongly the computation. e.g. delta = 100 vs.delta = 0.01

**We agree that this part was unclear in the first submission and thank the reviewer for highlighting that the parameters were not properly discussed.**

4. I am not very sure I fully understand it. But I will leave it to other reviewers!

**Maybe it is clearer with an (admittedly) extreme example: if there is some part of the domain where there are no training data and this domain is dynamically completely disconnected from the rest, then its value is (per construction) completely unconstrained, except for an a priori information of reasonable values. So e.g. 10°C is probably as good a guess as 14°C. The neural network tends to oscillate between these two values because there are no constraints from the**

| | last reconstruction | average reconstruction |
|---|---|---|
| missing data change every epoch | 0.4968 | 0.3834 |
| missing data change every 10 epochs | 0.4423 | 0.4146 |
| missing data change every 20 epochs | 0.4387 | 0.3984 |

**Table 1.** CV error for different experiments keeping the mask of the missing data constant for several epochs

**data. Assume further that we have a validation data point of 13°C in this area, then the average RMS would be (abs(10-13) + abs(14-13))/2 = 2 °C but the RMS of the average is abs(12-13) = 1 °C. (for a single value the RMS is directly related to the absolute value which we use here to simplify the notation, but the same results are true for a series of numbers).**

5. I would guess the fundamental reason why the RMSE and Loss fluctuate so much is that the random mark missing data in every mini batch. Because in every epoch, the spatial correlation of missing and available data is disrupted due to random marking, hence what the network has learned in previous epoch is disrupted as well, which eventually is reflected in RMSE and Loss. The fluctuations may not have so much to do with mini-batch optimization. Perhaps one way to check is to use same random mark missing data for every 20 epochs, and average at every 20 epochs. Just my opinion

**We conducted this experiment and the reviewer is right that it had indeed a quite significant/dramatic effect on the convergence of the cost function (see the attached figure). Unfortunately, the average reconstructed SST or the reconstruction from the last epoch are not better than the best experiments that we had already in the manuscript.**

**The experiment "missing data change every epoch" is the experiment "DINCAE**

(all skip connections and average pooling)" from the manuscript. Despite that the results are not better in this case, the reviewer's idea is promising and we include it as an option in our code. Application to other cases will tell if this proposed option (keeping the same data marked as missing for every 20 mini-batch) should rather be preferred.

Another way to interpret the marking of some data as missing would be to view it as a drop-out layer as the value of zero does indeed represent an "infinitely" large error. Change the mask of missing data at every epoch seems to help the generalization.

We also verified that the current version of our code reproduces exactly the same results as the code when the article was submitted if the same random seeds are used to exclude the possibility that any other change to the code has an impact here.
* * *
[Figure]

**Fig. 1.** Cost function when changing the data marked as missing change every 10 or 20 epochs

---

## Short Comment (SC1) · 1 Jan 2020

General Comments: This is a good work. The author develops a new reconstruction methodology based on neural network with the structure of a convolutional auto-encoder. In this manuscript, the author uses the cross-validation and field comparison to prove the performance of DINCAE in SST reconstruction is better than DINEOF from some statistical results, such as RMSE and standard deviation. I also use the DINCAE 1.0 released on GitHub by author to fill gapped chlorophyll satellite images in another sea area and achieved satisfactory results. The first reviewer put forward very meaningful comments, which promotes the improvement of this manuscript. I think there are

still some place to improve.

DINCAE is based on neural network, therefore its trained model can be saved and used to predict the other independent missing data. Considering the time parameter of DINCAE input is "days in year", the trained model should be able to fill the data for other years. I think this is one of the biggest different between DINCAE and DINEOF, but is not mentioned in this manuscript.

Many of the colorbar labels are incomplete or uninformative.

Page 7 line 1: Manuscript gives the size of complete dataset is 8 x 112 x 112 x 5266. But, I think "Scaled SST anomalies and inverse of error variance of the previous day" in the input parameters list (page 6 line 20) means two parameters and the size of complete dataset should be 10 x 112 x 112 x 5266.

Page 9 Equation 8: lacks a left bracket.

Page 11 line 1: Calculation efficiency is an important index to evaluate the reconstruction algorithm. The manuscript gives "Training this network for 1000 epochs takes 32 hours on a GeForce GTX 1080 and Intel Core i7-7700 with the neural network library tensorflow (Abadi et al., 2015)". I am also interesting on how many hours spent by DINEOF.

Page 12. Figure 3: What is the colorbar meaning? Please adding some information about colorbar in figure or the caption under figure.

Page16 line 5: The comparison between the measured data and the reconstructed data is the most persuasive validation method. I suggest adding the number of measured points, which can be more convincing.

Page18. Figure 7: The standard deviation of reconstructed SST by DINCAE and DINEOF along the southern coast of France and Corsica is larger than that of original SST. Why? The unit and label of colorbar are not given.

---

## Author Comment (AC3) · 8 Jan 2020

**We would like to thank the reviewer for carefully reading the manuscript and providing comments to the manuscript and to the figures in particular to improve their clarity. We hope that our reply below answers the questions and comments by the reviewer in an adequate manner. Our response is in bold-face.**

General Comments: This is a good work. The author develops a new reconstruction methodology based on neural network with the structure of a convolutional autoencoder. In this manuscript, the author uses the cross-validation and field comparison

to prove the performance of DINCAE in SST reconstruction is better than DINEOF from some statistical results, such as RMSE and standard deviation. I also use the DINCAE1.0 released on GitHub by author to fill gapped chlorophyll satellite images in another sea area and achieved satisfactory results. The first reviewer put forward very meaningful comments, which promotes the improvement of this manuscript. I think there are still some place to improve.

DINCAE is based on neural network, therefore its trained model can be saved and used to predict the other independent missing data. Considering the time parameter of DINCAE input is "days in year", the trained model should be able to fill the data for other years. I think this is one of the biggest different between DINCAE and DINEOF, but is not mentioned in this manuscript.

Many of the colorbar labels are incomplete or uninformative.

**We have added to all captions that the temperature and its standard deviation are in degrees Celsius and improved the caption for figure 3 (numbering of the original manuscript; figure 4 for the revised manuscript) as explained below.**

**We also clarified that the day of the year is not used by DINEOF by adding the following:**

> **It should be noted that DINEOF does not use the day of the year of each satellite image, but it uses a temporal filter which increases the temporal coherence of the reconstruction (Alvera-Azcárate et al., 2009).**

Page 7 line 1: Manuscript gives the size of complete dataset is 8 x 112 x 112 x 5266.But, I think "Scaled SST anomalies and inverse of error variance of the previous day"in the input parameters list (page 6 line 20) means two parameters and the size of complete dataset should be 10 x 112 x 112 x 5266.

**Thank you for pointing this out. This is now corrected. See also my reply to reviewer 1 from 17 December 2019.**

Page 9 Equation 8: lacks a left bracket.

**This is equation (8) from the original manuscript. We do not see where there is a missing bracket.**

$$p(y_{ij}|\hat{y}_{ij}, \hat{\sigma}_{ij}) = \frac{1}{\sqrt{2\pi\hat{\sigma}_{ij}^2}} \exp\left(-\frac{(y_{ij} - \hat{y}_{ij})^2}{2\hat{\sigma}_{ij}^2}\right) \tag{1}$$

**But we noticed a bracket-issue in equation (5) on page 8 which we corrected.**

Page 11 line 1: Calculation efficiency is an important index to evaluate the reconstruction algorithm. The manuscript gives "Training this network for 1000 epochs takes 32hours on a GeForce GTX 1080 and Intel Core i7-7700 with the neural network library tensorflow (Abadi et al., 2015)". I am also interesting on how many hours spent by DINEOF.

**The original version of DINCAE took indeed 32 hours to make 1000 epochs, but after restructuring the code (and some update of system libraries and drivers, like the NVIDIA kernel driver), the version now published on github uses only 4.5 hours. These differences are indeed quite significant, but we are not able to tell which change caused this difference in runtime (the results are still the same). As a comparison, DINEOF uses 15 hours on an Intel Core i7-3930K CPU. Note also that the i7-3930K (used by DINEOF) was first released in 2011 and the i7-7700 (used by DINCAE) was released in 2017. DINCAE used thus a more modern CPU than DINEOF and the runtimes are not directly comparable. The manuscript is updated accordingly.**

Page 12. Figure 3: What is the colorbar meaning? Please adding some information about colorbar in figure or the caption under figure.

**In the manuscript we mentioned: "The color represents the estimated expected error standard deviation of the reconstruction" but we agree that it is more readable if this information is also placed in the caption. In the revised manuscript, this information is now also added to the caption. We thank the reviewer for pointing this out.**

Page16 line 5: The comparison between the measured data and the reconstructed data is the most persuasive validation method. I suggest adding the number of measured points, which can be more convincing.

**I assume that the reviewer refers here to the in situ measurements (as the number of validation data points measured by the satellite is already mentioned in the manuscript). For the in situ data, there are 774 surface observations. We agree that this is a useful information for the reader and it has been added to the manuscript.**

Page18. Figure 7: The standard deviation of reconstructed SST by DINCAE and DINEOF along the southern coast of France and Corsica is larger than that of original SST. Why? The unit and label of colorbar are not given.

**Near the coast there are much fewer observations than in the open ocean. We can illustrate this by comparing two different locations: the nearshore location at 8.0641°E and 43.9243°N with a standard deviation of 0.85 °C (around the seasonal cycle and the original SST) and the offshore location 8.0641°E and 43.0454°N, same longitude but different latitude with a standard deviation of 1.20 °C. There are about 9 times more data in the offshore location than in the**

**nearshore location (3037 versus 346). Also the data available in the nearshore is not evenly distributed around the different years (see the attached histograms) while the distribution is quite even for the offshore location. When the seasonal mean is computed from the available data it is thus biased towards the years where most data are available which results in a low standard deviation. Per definition, the data from DINCAE and DINEOF are complete time series and such distribution bias is not present.**

**The figure's caption now includes the units (degrees Celsius).**

**References**

Alvera-Azcárate, A., Barth, A., Sirjacobs, D., and Beckers, J.-M.: Enhancing temporal cor-
relations in EOF expansions for the reconstruction of missing data using DINEOF, Ocean
Science, 5, 475–485, https://doi.org/10.5194/os-5-475-2009, http://www.ocean-sci.net/5/475/
2009/, 2009.

Number of obs. per year at 8.0641°E and 43.9243°N

**Fig. 1.** Data histogram per year for a nearshore location

Interactive
comment

[Figure]

Number of obs. per year at 8.0641°E and 43.0454°N

**Fig. 2.** Data histogram per year for a offshore location

---

## Referee Comment (RC4) · Anonymous Referee #2 · 8 Feb 2020

Review of DINCAE 1.0: a convolutional neural network with error estimates to reconstruct sea surface temperature satellite observations

This study presents a novel approach of reconstructing sea surface temperatures from cloudy satellite data by making good use of modern deep learning techniques. While I believe that the study has been carefully designed and executed, I have severe problems with the paper in terms of its presentation and accuracy in writing. This study could become a high-impact publication if it were better structured and methods and outcomes were described clearer. I advise the authors to apply major revisions and seek the help of a native speaker to avoid erroneous or ambiguous statements. Below,

please find my detailed comments.

Abstract: the sentence "However, it is unclear how to handle missing data (or data with variable accuracy) in a neural network when using incomplete satellite data in the training phase." is not very clear. Perhaps rephrase as "Contrary to standard image reconstruction with neural networks, this application requires a method to handle missing data (or data with variable accuracy)."

L7: suggest to remove "essentially"

L9: what is "relatively long"? Provide a number, please.

L11: "a method to reconstruct missing data": suggest to rephrase "a previously published method", "the current standard method", "the state-of-the-art DINEOF method", or similar.

L16: what is meant by "the ocean current signal"? A signal always refers to a measurement or sensing process. Here you want to refer to a physical process in the ocean.

L17: replace "like" with "e.g."

L20: replace "sensor" with "measurement". This sentence refers to the measurement principle, not the technical instrument, which performs the measurement.

L22: "but often small scale information is filtered out because of the transient and stochastic nature of these structures." The transient and stochastic nature produces variability and is clearly not the reason why small scale information is filtered out. This is rather a result of the averaging procedures which are applied in practically all known techniques to interpolate. As this is a critical sentence for setting the stage of this study, please rethink the phrasing and provide a more precise description of the issue, which you are trying to solve.

L23: DINEOF falls from the sky here. For non-experts in the field of sea surface temperature reconstructions, it is completely unclear what this is. Also, as DINEOF

appears here for the first time, it is a must that you provide a reference. The reference comes two sentences further down, which is too late. A brief description of the method would be appropriate here.

page 2: L9: "to detect the presence of non-linear, stochastic features" - I disagree that neural networks "detect" these features. Rather they are able to "learn" such features and thus potentially produce more detailed reconstructions of them.

L11: This statement is too general. I strongly suggest to first explain briefly what types of neural networks exist and how they can/could be used for the problem you want to solve (including references to the most important deep learning papers). Then you can make the argument that these networks (and in particular CNN derivates) are generally trained with complete data, whereas in your application you need to find a method which can train with scenes containing missing data, because there are no complete satellite scenes available (or only very few).

L14: this paragraph contains some of the literature review I am asking for in my previous comment. However, here it is on the one hand too specific (only ocean data applications), but on the other hand too superficial as it doesn't become clear why you need to develop a new approach and cannot simply apply for example the method of Krasnopolsky et al.

L33: I storngly suggest to re-organize the paper so that it follows the classical structure and describes the method before the data, and in particular before another reconstruction (DINEOF) is reported.

Page 3: L4: Delete the first sentence. You don't need a motivation within your "Data" section. Such an argument belongs in the introduction, if you wish to explain, for example, why you designed your study based on this dataset and not another one. In section 2 you should only describe the dataset, without any "discussion".

L19: The cross-validation deserves more explanation, because you are publishing in

a journal which is primarily read by non-experts in the field of machine learning. It is important to note that (contrary to standard image analysis) subsequent scenes from the AVHRR data are not independent. Therefore you cannot use a random sample to construct your test dataset (or "validation dataset", whichever terminology you prefer). I am wondering if 50 scenes are indeed sufficient to thoroughly test the generalization of the network. Not being an ocean scientist, I can only assume that typical transport time scales in your study region are on the order of a week(?). This would imply that the first 7 scenes of your "independent" test data are still "polluted" and thus not fully independent. Have you tried retaining a larger test sample?

L21: How can you retain > 100,000 measurements from 50 scenes? Are these individual pixels, or did you actually apply cross-validation with random sampling, thus ignoring the argument I made above?

Page 4: Figure 1: I cannot see any arrows, which are referenced in the figure caption.

L3: Again, some explanation of DINEOF is warranted in this paper. It should be clear what this method does, without having to access the referenced papers. For details you can refer to them, but not for the fundamental "explanation".

Page 5: Table 1: instead of "fewer layers" or "more layers", the number of layers should be given.

L5: I don't think this is an appropriate citation here. It is the principle of EOFs to detect relations between variables and construct an orthogonal set of linear functions to model these relations. Due to the cutoff after N EOFs, there is always smoothing applied. A proper citation here would be some standard statistics book.

L7: I am not at all surprised by this result: if you extend the timeseries, you will be more likely to sample patterns, which have not been observed before and which don't fit well to the already "learned" EOFs. Hence, there is less structure that can be described by the EOFs and more noise.

L14: I disagree that deep neural networks are "extensively" used in Earth sciences. This field is developing rapdily, but the applications are so far far from "extensive".

Page 6: L6: what does "different errors" mean? Different to what? Also: as mentioned before, for an article in this journal there needs to be a brief description of CAEs in the method section, which should contain enough information that an uninitiated reader (e.g. an ocean modeller) understands why the approach might actually work. Clearly, before the discussion taking place here, the reader must know how the network is constructed, which activation functions are used, which optimizer is used, whether regularization techniques are applied, etc. And the basis for the network built here is probably coming from some deep learning paper, which then needs to be cited. Such a description follows on page 7. Please restructure.

Page 7: L7: the references refer to convolutional layers only. However, you are applying an autoencoder approach, so the appropriate references should be made. For example: G.E. Hinton and R.R. Salakhutdinov. Reducing the dimensionality of data with neural networks. Science, 313(5786):504, 2006.

L9: "different numbe rof filters" - not filter sizes. Your filter size is always 3x3 as you state-of-the-art just below.

L16: composed of

Page 8: Please also write down the loss function.

Page 9: L2: So here it appears that indeed your "independent" validation data are not independent.

L9: I don't understand the random masking: is one random mask applied to each image and then the same image used in each epoch? Or do you apply different random masks to the same image as a means to augment your data and increase generalization?

L24: Remove the statement about other variables, because you focus exclusively on SST. This can go in the conclusions section, but is confusing here.

[Figure]

Page 10: L10: I am confused here as to how the reconstruction and the training works. Normally, you first train your network and then you reconstruct (in particular on unseen data). Then you cannot take any average between epochs 200 and 1000.

Page 13: L5: "[..] underestimate the actual error by 15% but one can argue that an underestimation of the expected error of this magnitude should be acceptable for most purposes." This sentence deserves further explanation. What is the generally accepted accuracy of the error estimate?

Page 14: L4: again: not "filter sizes" but "number of filters"

Page 15: Figure 5: the figure titles are misleading. Apparently you are always showing results for one specific day. This day should then be mentioned in the caption and nt as title on the first panel. The way the panels are labelled now suggests that you compare apples with oranges (which I don't believe you do). Also here and in Figure 6 it is not quite clear to me if the DINEOF reconstruction also had to deal with the "added clouds" or not. Higher up, when you mention the addition of random clouds it would be good to see a typical fraction of image size which is obstructed by these random clouds. From figures 5 and 6, this obstruction seems to be quite large.

Page 17: L4: where can the reader see the comparison between in-situ obs and the reconstructions? This paragraph remains qualitative and doesn't contribute anything meaningful.

L15: indeed - if the deep learning method was applied correctly (which is somewhat difficult to judge from this paper due to all the issues described in this review), then this is a very nice and important result, which shows the superiority of deep learning approaches with their ability to learn non-linear functions compared to standard statistical methods. This key result could probably be brought out even clearer.

Page 18: L2: why is this method "practical"? I assume it is, but this is only because of my background knowledge. This point needs to be made explicit somewhere in the

paper. You list computation times for training, but you don't say how long it takes to reconstruct a scene once the network has been trained.

Page 19: After rewriting other parts of the manuscript I suggest to re-read the final part of the conclusions to see if the paper ends with the highest impact message.

---

## Author Comment (AC4) · 25 Feb 2020

Reply to the review of DINCAE 1.0: a convolutional neural network with error estimates to recon- struct sea surface temperature satellite observation

**We would like to thank the reviewer for carefully reading the manuscript. Her/His complete review is copied below while our answers are inserted below every comment and written in boldface.**

This study presents a novel approach of reconstructing sea surface temperatures from cloudy satellite data by making good use of modern deep learning techniques. While I

believe that the study has been carefully designed and executed, I have severe problems with the paper in terms of its presentation and accuracy in writing. This study could become a high-impact publication if it were better structured and methods and outcomes were described clearer. I advise the authors to apply major revisions and seek the help of a native speaker to avoid erroneous or ambiguous statements. Below, please find my detailed comments.

**All coauthors and I, tried our best to improve the manuscript to avoid erroneous or ambiguous statements. We hope that the manuscript is now clear.**

Abstract: the sentence "However, it is unclear how to handle missing data (or data with variable accuracy) in a neural network when using incomplete satellite data in the training phase." is not very clear. Perhaps rephrase as "Contrary to standard image reconstruction with neural networks, this application requires a method to handle missing data (or data with variable accuracy)."

**OK, we changed this sentence to:**

> **Contrary to standard image reconstruction with neural networks, this application requires a method to handle missing data (or data with variable accuracy) already in the training phase.**

L7: suggest to remove "essentially"

**Ok, done.**

L9: what is "relatively long"? Provide a number, please.

**We agree, and we changed the sentence to:**

**The approach, called DINCAE (Data-Interpolating Convolutional Auto-Encoder) is applied to a 25-year time-series of Advanced Very High Resolution Radiometer (AVHRR) sea surface temperature data**

L11: "a method to reconstruct missing data": suggest to rephrase "a previously published method", "the current standard method", "the state-of-the-art DINEOF method", or similar.

**Thank you for the suggestion. In the revised manuscript we refer to the methods as "state-of-the-art".**

L16: what is meant by "the ocean current signal"? A signal always refers to a measurement or sensing process. Here you want to refer to a physical process in the ocean.

**We used the term signal indeed quite broadly in the original manuscript and replaced it in the revised manuscript with a more precise term. In the revised manuscript, this was changed to "the ocean velocity variability depends thus partially on ocean temperature".**

L17: replace "like" with "e.g."

**OK, done.**

L20: replace "sensor" with "measurement". This sentence refers to the measurement principle, not the technical instrument, which performs the measurement.

**The submitted manuscript reads:**

**However, as for any sensor working in the infrared or visible bands, clouds often obscure large parts of the field-of-view.**

**In the revised manuscript we changed sensor by "measuring technique".**

L22: "but often small scale information is filtered out because of the transient and stochastic nature of these structures." The transient and stochastic nature produces variability and is clearly not the reason why small scale information is filtered out. This is rather a result of the averaging procedures which are applied in practically all known techniques to interpolate. As this is a critical sentence for setting the stage of this study, please rethink the phrasing and provide a more precise description of the issue, which you are trying to solve.

**We agree. The following has been added to the manuscript:**

> **A truncated EOF decomposition will focus primarily in spatial structures with a "strong" signature (or more formally defined with a significant L2 norm compared to the total variance). Small scale structures can be included in a truncated EOF decomposition as long as their related variance is large enough to be present in the retained EOF modes. But small scale structures tend to be transient (short-lived) and therefore are often not retained in the dominant EOF modes. It should be noted that there is no explicit spatial filtering scale in DI-NEOF removing small scales (unlike other methods like optimal interpolation, kriging, spline interpolation). But in practice a similar smoothing effect is noticed because of the EOF truncation.**

**We removed the terms "transient" and "stochastic" in the revised manuscript and we clarified that there is no explicit filtering in DINEOF by using a predefined spatial length-scale.**

L23: DINEOF falls from the sky here. For non-experts in the field of sea surface temperature reconstructions, it is completely unclear what this is. Also, as DINEOF appears here for the first time, it is a must that you provide a reference. The reference comes two sentences further down, which is too late. A brief description of the method would be appropriate here.

**The original manuscript was:**

> **DINEOF (Data Interpolating Empirical Orthogonal Functions), provides an accurate way of retrieving missing data and reducing noise in satellite datasets using a set of optimal EOFs. The optimal number of EOF is determined by cross-validation. More information on the DINEOF approach is documented in (Beckers and Rixen, 2003;Alvera-Azcárate et al., 2005).**

**The reference was now put in the first sentence. The following information was added when describing DINEOF. More information has been added later in DINEOF sections, because it would be too technical for the introduction.**

> **DINEOF (Data Interpolating Empirical Orthogonal Functions, Beckers and Rixen, 2003; Alvera-Azcárate et al., 2005), is an iterative method to reconstruct missing observations reducing noise in satellite datasets using empirical orthogonal functions (EOF). A truncated EOF decomposition using the leading EOFs is performed and the initially missing data are reconstructed using this EOF decomposition. The EOF decomposition and reconstruction is repeated until convergence.**

page 2: L9: "to detect the presence of non-linear, stochastic features" - I disagree that neural networks "detect" these features. Rather they are able to "learn" such features and thus potentially produce more detailed reconstructions of them.

**OK, this is changed the revised manuscript:**

> **Neural networks are therefore specially well positioned to learn non-linear, stochastic features measured at the sea surface by satellite sensors, and their use might prove efficient in retaining these structures when analysing satellite data, for example for reconstructing missing data.**

L11: This statement is too general. I strongly suggest to first explain briefly what types of neural networks exist and how they can/could be used for the problem you want to solve (including references to the most important deep learning papers). Then you can make the argument that these networks (and in particular CNN derivates) are generally trained with complete data, whereas in your application you need to find a method which can train with scenes containing missing data, because there are no complete satellite scenes available (or only very few).

**We added the following information to the manuscript to give a general overview of the types of neural networks. Later in the manuscript we will focus on some examples in oceanography.**

> **Neural networks can be composed of a wide variety of building blocks, such as fully connected layers (Rosenblatt, 1958; Widrow and Hoff, 1962) recurrent networks (e.g. Long Short-term Memory (Hochreiter and Schmidhuber, 1997), Gated recurrent unit (Cho et al., 2014)), convolutional layers (LeCun et al., 1998; Krizhevsky et al., 2012). Recurrent networks work typically with a one dimensional list of inputs of a variable length (such as a text sentence). Fully connected layers and convolutional layers require to have a full dataset without missing**

**data, at least for the training phase. For a review on neural networks the reader is referred to Schmidhuber (2015) and references therein.**

L14: this paragraph contains some of the literature review I am asking for in my previous comment. However, here it is on the one hand too specific (only ocean data applications), but on the other hand too superficial as it doesn't become clear why you need to develop a new approach and cannot simply apply for example the method of Krasnopolsky et al.

**In the revised manuscript we first referenced the general idea in artificial neural networks and then give some examples in oceanography. The field is too wide to give a comprehensive overview of all applications in geoscience and we limit therefore the applications to oceanography. The updated manuscript goes into more details on the limitations of the method of Krasnopolsky et al. (2016):**

> **The neural network by Krasnopolsky et al. (2016) uses as input satellite sea surface elevation, sea surface salinity, sea surface temperature and *in situ* Argo salinity and temperature vertical profiles with some auxiliary information (like longitude, latitude and time) to estimate the Chlorophyll-a concentration. The network does not use measured Chlorophyll-a concentration at a given location as input during inference (the reconstruction phase), nor the information from nearby grid points to infer Chlorophyll-a concentration. The network is exposed to the chlorophyll measurements only during the training phase.**

L33: I storngly suggest to re-organize the paper so that it follows the classical structure and describes the method before the data, and in particular before another reconstruction (DINEOF) is reported.

**For us it was important to describe the data first because the ubiquity of clouds and the strong seasonal cycle in the data sets are an important constraint for the method. Therefore we believe that for most readers it will be easier to understand the method once its input (i.e. the satellite data) is presented. In fact, the type of data motivates the choices that were made during the design of the method. Also, we believe it helps the reader if we give concrete size of the arrays and matrices involved when the method is described. However, these matrix sizes depend on the data. Since the method section depends heavily on the data section, and the data section does not depend on the method section, we choose to present the data first. We actually tried to reverse the order as suggested by the reviewer but we ended up with too many forward references which would harm the readability.**

**We however presented the DINEOF method after the DINCAE method as suggested in the revised manuscript.**

Page 3: L4: Delete the first sentence. You don't need a motivation within your "Data" section. Such an argument belongs in the introduction, if you wish to explain, for example, why you designed your study based on this dataset and not another one. In section 2 you should only describe the dataset, without any "discussion".

**We deleted the first sentence.**

L19: The cross-validation deserves more explanation, because you are publishing in a journal which is primarily read by non-experts in the field of machine learning. It is important to note that (contrary to standard image analysis) subsequent scenes from the AVHRR data are not independent. Therefore you cannot use a random sample to construct your test dataset (or "validation dataset", whichever terminology you prefer). I am wondering if 50 scenes are indeed sufficient to thoroughly test the generalization of the network. Not being an ocean scientist, I can only assume that typical transport

time scales in your study region are on the order of a week(?). This would imply that the first 7 scenes of your "independent" test data are still "polluted" and thus not fully independent. Have you tried retaining a larger test sample?

**In the revised manuscript we also added a reference to a classical textbook for the cross-validation method outside of the realm of machine learning.**

> **To assess the accuracy of the reconstruction method, cross-validation is used wilks1995. For cross-validation a subset of the data is withheld from the analysis and the final reconstruction is compared to the withheld dataset to access its accuracy. Since clouds have a spatial extent, we wanted to withhold data with a similar spatial structure. In the last 50 images we removed data according to the cloud mask of the first 50 images of the SST time series. The last 50 images represent the data from 2009-09-25 to 2009-12-27 (since some scenes with too few data have been dropped as mentioned before). These data are not used at all during either the training or the reconstruction phases, and can therefore be considered independent. In total, 106 816 measurements (i.e. individual pixels) have been withheld this way.**

**We did not try to have a temporal gap between the training data and test data or a larger test sample. It is true that there is some correlation between the training data and test data (the last few scenes used for learning might correlate with the first few scenes of the validation set). But we also performed a validation with in situ data in the manuscript. Both validation methods lead to the same outcome. We realize by reading the other comments from the reviewer that the reference to the table with in situ validation was not quite clear in the original manuscript.**

**For the "best" DINCAE experiment, we also recomputed the cross-validation**

**RMS error using only the last 43 scences and the RMS error is with 0.3754 °C very similar (and even slightly lower) that the RMS error using the last 50 images 0.3834°C. If there would be a significant "pollution" effect, then one would expect that the RMS error with 43 scence to be larger than with the 50 scences. But this was not observed. Given the large pool of training data (5216 scenes) the effect of a handful potentially correlated scenes does not have any significant effect.**

L21: How can you retain > 100,000 measurements from 50 scenes? Are these individual pixels, or did you actually apply cross-validation with random sampling, thus ignoring the argument I made above?

**Yes, this is the count of individual pixels of the 50 images used for cross-validation as measured by the AVHRR sensor. We clarified this in the revised manuscript.**

Page 4: Figure 1: I cannot see any arrows, which are referenced in the figure caption.

**We are sorry about this problem. It has been corrected in the revised manuscript.**

L3: Again, some explanation of DINEOF is warranted in this paper. It should be clear what this method does, without having to access the referenced papers. For details you can refer to them, but not for the fundamental "explanation".

**We expand this section and included more information about how DINEOF reconstructed the missing data:**

**A truncated EOF decomposition using the leading $N$ EOFs is performed and the initially missing data are reconstructed by combining the retained EOF modes and their corresponding amplitudes. The**

**EOF decomposition and reconstruction of missing data is repeated until convergence. The optimal number of EOFs $N$ is determined by cross-validation.**

Page 5: Table 1: instead of "fewer layers" or "more layers", the number of layers should be given.

**We agree, and this has been added to the table ("fewer layers": 3 convolutional layers and "more layers": 5 convolutional layers)**

L5: I don't think this is an appropriate citation here. It is the principle of EOFs to detect relations between variables and construct an orthogonal set of linear functions to model these relations. Due to the cutoff after N EOFs, there is always smoothing applied. A proper citation here would be some standard statistics book.

**We added a reference to Wilks (1995).**

L7: I am not at all surprised by this result: if you extend the timeseries, you will be more likely to sample patterns, which have not been observed before and which don't fit well to the already "learned" EOFs. Hence, there is less structure that can be described by the EOFs and more noise

**Original manuscript reads:**

**As only 13 modes are retrained by DINEOF for the reconstruction, some small scale structures are smoothed-out, which is a well known property of a truncated EOF decomposition (e.g. Alvera-Azcárate et al., 2009). This smoothing effect results in an RMS error of 0.3864°C**

**when comparing the reconstructed dataset to all the initially present SST (i.e. used for the reconstruction). A somewhat surprising result is that when using less data (only from the last two years, i.e. 2008 to 2009), 19 EOFs modes are retained, leading to a reconstruction with richer structures.**

Here we did not extend the time series but just used a subsample of the time series. For the full time series (1985-2009, 25 years), 13 modes have been retained as optimal. For a subset (2008-2009, 2 years), more EOFs (19 modes) have been retained. The number of time instances is an upper bound for the number of EOFs with non-zero singular values. For a shorter time series, this upper bound is thus lower, yet more EOFs have been retained with the shorter time series. This result was unexpected for us.

L14: I disagree that deep neural networks are "extensively" used in Earth sciences. This field is developing rapdily, but the applications are so far far from "extensive".

**We agree and the sentence was revised:**

> **Convolutional and other deep neural networks are extensively used in computer vision and find an increasing number of applications in Earth sciences [. . .]**

Page 6: L6: what does "different errors" mean? Different to what?

**The error can be different from one pixel to another. We changed this in the revised manuscript as "error varying in space and/or time" to be more clear.**

Also: as mentioned before, for an article in this journal there needs to be a brief description of CAEs in the method section, which should contain enough information that an uninitiated reader (e.g. an ocean modeller) understands why the approach might actually work. Clearly, before the discussion taking place here, the reader must know how the network is constructed, which activation functions are used, which optimizer is used, whether regularization techniques are applied, etc. And the basis for the network built here is probably coming from some deep learning paper, which then needs to be cited. Such a description follows on page 7. Please restructure.

**This is the structure of the original manuscript:**

**On page 6:**

- **We explain how missing data is handled in data assimilation which motivates the present work**

- **Handling of missing data in the input is done in analogy here**

- **Describe the input of the neural network**

**On page 7-8:**

- **General structure of the network**

- **Skip connections**

- **activation function**

**On page 9:**

- **Cost function**

**On page 10:**

- **Optimizer**

- **regularization techniques**

**To us, it seems logical and didactical to make a description of the method step-by-step: first the input of the network, then how the input is transformed by the network (general structure, activation function, skip connections), how to assess the accuracy of the output (cost function), how to optimize the accuracy and finally how to prevent overfitting (regularization techniques). In the revised manuscript we make it clear that the description of the convolutional autoencoder will come in the following.**

**In the revised manuscript, we put the reference to Hinton and Salakhutdinov (2006); Ronneberger et al. (2015) more prominently as the neural network structure proposed in those papers is quite similar to one used here. We also updated the overview of the paper in the introduction to make the links between sections clearer.**

Page 7: L7: the references refer to convolutional layers only. However, you are applying an autoencoder approach, so the appropriate references should be made. For example: G.E. Hinton and R.R. Salakhutdinov. Reducing the dimensionality of data with neural networks. Science, 313(5786):504, 2006.

**Note that we cite Hinton and Salakhutdinov (2006) on page 2 in the introduction of the original manuscript:**

> **An auto-encoder is a particular type of network which can compress and decompress the information in an input dataset (Hinton and**

Salakhutdinov, 2006), effectively reducing the dimensionality in the
input data.

**To put the reference of Hinton and Salakhutdinov (2006) more prominently in the
manuscript we changed on page 7, line 7 of the revised manuscript:**

**The main building blocks of the neural network (Table 2) are convolu-
tional layers (LeCun et al., 1998; Krizhevsky et al., 2012).**

**was changed to:**

**The overall structure of the neural network (Table 2) is a convolutional
autoencoder (Hinton and Salakhutdinov, 2006; Ronneberger et al.,
2015). Its main building blocks are convolutional layers (LeCun et al.,
1998; Krizhevsky et al., 2012).**

L9: "different number of filters" - not filter sizes. Your filter size is always 3x3 as you
state-of-the-art just below.

**Thank you for pointing this out. We corrected this and changed it throughout the
manuscript.**

L16: composed of

**Thank you, we corrected this (and we found a similar error which is corrected
too).**

Page 8: Please also write down the loss function.

**The cost function is "Equation 9" (page 9) from the original manuscript:**

**[...] The cost function has finally the following form:** $\mathbf{J}(\hat{y}_{ij}, \hat{\sigma}_{ij}) =$
$\frac{1}{2N} \sum_{ij} \left[ \left( \frac{y_{ij} - \hat{y}_{ij}}{\hat{\sigma}_{ij}} \right)^2 + \log(\hat{\sigma}_{ij}^2) + 2\log(\sqrt{2\pi}) \right]$

**We do not make a distinction between "cost function" and "loss function" and use it as synonyms as in Goodfellow et al. (2016):**

> **The function we want to minimize or maximize is called the objective function, or criterion. When we are minimizing it, we may also call it the cost function, loss function, or error function. (page 80, of Goodfellow et al. (2016))**

**The loss function per individual scalar sample is the term in brackets of equations 9. This information has been added to the manuscript.**

Page 9: L2: So here it appears that indeed your "independent" validation data are not independent.

**This is line 2, page 9 for the original manuscript:**

> **The input data set is randomly shuffled (over the time dimension) and partitioned into so-called mini-batches of 50 images, as an array of the size 8 x 112 x 112 x 50.**

**It is unclear to us why the reviewer thinks that the data is not independent. The data marked for cross-validation is not used during the training. It is just a coincidence that the mini-batch size is equal to the number of images used for cross-validation. These numbers are not related.**

L9: I don't understand the random masking: is one random mask applied to each image and then the same image used in each epoch? Or do you apply different random masks to the same image as a means to augment your data and increase generalization?

**The later is the case, the paragraph has been revised to make this clear:**

> **For every input image, more data points were masked (in addition to the cross-validation) by using a randomly chosen cloud mask during training. The cloud mask of a training image would thus be the union of the cloud mask of the input dataset and a randomly chosen cloud mask. This allows us to assess the capability of the network to recover missing data under clouds. Without the additional clouds, the neural network would simply learn to reproduce the SST values that are already received as input. At every epoch a different mask is applied to a given image to mitigate overfitting and aid generalization.**

L24: Remove the statement about other variables, because you focus exclusively on SST. This can go in the conclusions section, but is confusing here.

**OK, done.**

Page 10: L10: I am confused here as to how the reconstruction and the training works. Normally, you first train your network and then you reconstruct (in particular on unseen data). Then you cannot take any average between epochs 200 and 1000.

**The revised paragraph now reads:**

> **The neural network is updated using the gradient for every mini-batch during training and after every 10 epochs the current state of the neu-**
[Figure]

**ral network is used to infer the missing data over the whole time series, and in particular reconstructing the missing data is the cross-validation dataset. But importantly, the network is not updated using the cross-validation data.**

**So effectively, we temporarily suspend the training after every 10 epochs and reconstruct the missing data but then continue the training.**

Page 13: L5: "[..] underestimate the actual error by 15underestimation of the expected error of this magnitude should be acceptable for most purposes." This sentence deserves further explanation. What is the generally accepted accuracy of the error estimate?

**In the revised manuscript we avoided the term "acceptable" and added the following information:**

> **An interpolation technique which is commonly used in operational context, is optimal interpolation. This technique is able to provide an expected error variance of the interpolated fields based on a series of assumptions, in particular that the errors are Gaussian distributed with a known covariance and zero mean. Given these assumptions, the error variance of the optimal interpolation algorithm is only found to be weakly related to the observed RMSE in a study of Pisano et al. (2016) using satellite sea-surface temperature in the Mediterranean Sea. In this context, the fact that DINCAE underestimates the actual error only by 15% on average can be seen as an improvement.**

Page 14: L4: again: not "filter sizes" but "number of filters"

**Thank you, this is corrected. We corrected all 5 occurrences of this issue.**

Page 15: Figure 5: the figure titles are misleading. Apparently you are always showing results for one specific day. This day should then be mentioned in the caption and nt as title on the first panel. The way the panels are labelled now suggests that you compare apples with oranges (which I don't believe you do). Also here and in Figure 6 it is not quite clear to me if the DINEOF reconstruction also had to deal with the "added clouds" or not. Higher up, when you mention the addition of random clouds it would be good to see a typical fraction of image size which is obstructed by these random clouds. From figures 5 and 6, this obstruction seems to be quite large.

**We removed the dates from the first panel. It was already mentioned in the caption and the reviewer is right that the date is common to all panels of figure 5.**

**Initially, the averaged cloud coverage of the dataset is 46% (over all 25 years). The cloud coverage for the 50 last scenes is increased to 77% when the cross-validation points are excluded. It is true that a significant part of the scene is obscured (after marking the data for cross-validation), but in the Mediteranan Sea the cloud coverage is relatively low compared to the globally average cloud coverage which is 75% (Wylie et al., 2005). Removing some data for cross-validation makes the cloud coverage thus more similar to the global average.**

Page 17: L4: where can the reader see the comparison between in-situ obs and the reconstructions? This paragraph remains qualitative and doesn't contribute anything meaningful.

**The RMS errors can be seen in table 3, but the reference in the original manuscript was unfortunately not clear:**

> **As expected, biases play now a more important role when comparing**

**in situ observations with reconstructed satellite data (3).**

**"(3)" has been changed to "(Table 3)". We apologize for this issue which could easily cause a reader to overlook this table.**

L15: indeed - if the deep learning method was applied correctly (which is somewhat difficult to judge from this paper due to all the issues described in this review), then this is a very nice and important result, which shows the superiority of deep learning approaches with their ability to learn non-linear functions compared to standard statistical methods. This key result could probably be brought out even clearer.

**Thank you for your encouragement! We hope that the revised manuscript based on the comments of all reviewers is now clearer.**

Page 18: L2: why is this method "practical"? I assume it is, but this is only because of my background knowledge. This point needs to be made explicit somewhere in the paper. You list computation times for training, but you don't say how long it takes to reconstruct a scene once the network has been trained.

**Reconstructing the data of all 25 years takes only 8 seconds on the GeForce GTX 1080 for a trained network, but training the network can take several hours as mentioned in the manuscript. The manuscript has been updated with the reconstruction time for a trained network.**

**In the revised manuscript we removed the term "practical" because it was not possible for us to give it a precise meaning.**

Page 19: After rewriting other parts of the manuscript I suggest to re-read the final part of the conclusions to see if the paper ends with the highest impact message.

We agree that the ending of the conclusion was quite dull in the original manuscript. We revised the conclusions accordingly:

The tests conducted in this paper show that DINCAE is able to provide a good reconstruction of missing data in satellite SST observations and retaining more variability than the DINEOF method. In addition, the expected error variance of the reconstruction is estimated avoiding several assumptions (difficult to justify in practice) of other methods like optimal interpolation.

**References**

Alvera-Azcárate, A., Barth, A., Rixen, M., and Beckers, J.-M.: Reconstruction of incomplete oceanographic data sets using Empirical Orthogonal Functions. Application to the Adriatic Sea Surface Temperature., Ocean Modelling, 9, 325–346, https://doi.org/10.1016/j.ocemod.2004.08.001, http://hdl.handle.net/2268/4296, 2005.

Beckers, J.-M. and Rixen, M.: EOF calculation and data filling from incomplete oceanographic datasets, Journal of Atmospheric and Oceanic Technology, 20, 1839–1856, https://doi.org/10.1175/1520-0426(2003)020<1839:ECADFF>2.0.CO;2, 2003.

Cho, K., van Merrienboer, B., Gülçehre, Ç., Bougares, F., Schwenk, H., and Bengio, Y.: Learning Phrase Representations using RNN Encoder-Decoder for Statistical Machine Translation, CoRR, abs/1406.1078, http://arxiv.org/abs/1406.1078, 2014.

Goodfellow, I., Bengio, Y., and Courville, A.: Deep Learning, MIT Press, http://www.deeplearningbook.org, 2016.

Hinton, G. E. and Salakhutdinov, R. R.: Reducing the Dimensionality of Data with Neural Networks, Science, 313, 504–507, https://doi.org/10.1126/science.1127647, https://science.sciencemag.org/content/313/5786/504, 2006.

Hochreiter, S. and Schmidhuber, J.: Long Short-term Memory, Neural computation, 9, 1735–80, https://doi.org/10.1162/neco.1997.9.8.1735, 1997.

Krasnopolsky, V., Nadiga, S., Mehra, A., Bayler, E., and Behringer, D.: Neural Networks Technique for Filling Gaps in Satellite Measurements: Application to Ocean Color Observations, Computational Intelligence and Neuroscience, 2016, https://doi.org/10.1155/2016/6156513, 2016.

Krizhevsky, A., Sutskever, I., and Hinton, G. E.: ImageNet Classification with Deep Convolutional Neural Networks, in: Advances in Neural Information Processing Systems 25, edited by Pereira, F., Burges, C. J. C., Bottou, L., and Weinberger, K. Q., pp. 1097–1105, Curran Associates, Inc., http://papers.nips.cc/paper/4824-imagenet-classification-with-deep-convolutional-neural-networks.pdf, 2012.

LeCun, Y., Bottou, L., Bengio, Y., and Haffner, P.: Gradient-Based Learning Applied to Document Recognition, Proceedings of the IEEE, 86, 2278–2324, 1998.

Pisano, A., Nardelli, B. B., Tronconi, C., and Santoleri, R.: The new Mediterranean optimally interpolated pathfinder AVHRR SST Dataset (1982-2012), Remote Sensing of Environment, 176, 107 – 116, https://doi.org/10.1016/j.rse.2016.01.019, 2016.

Ronneberger, O., Fischer, P., and Brox, T.: U-Net: Convolutional Networks for Biomedical Image Segmentation, in: Medical Image Computing and Computer-Assisted Intervention – MICCAI 2015, edited by Navab, N., Hornegger, J., Wells, W. M., and Frangi, A. F., pp. 234–241, Springer International Publishing, Cham, https://doi.org/10.1007/978-3-319-24574-4_28, 2015.

Rosenblatt, F.: The perceptron: A probabilistic model for information storage and organization in the brain, Psychological Review, 65, 386–408, https://doi.org/10.1037/h0042519, 1958.

Schmidhuber, J.: Deep learning in neural networks: An overview, Neural Networks, 61, 85 – 117, https://doi.org/10.1016/j.neunet.2014.09.003, 2015.

Widrow, B. and Hoff, M. E.: Associative Storage and Retrieval of Digital Information in Networks of Adaptive "Neurons", pp. 160–160, Springer US, Boston, MA, https://doi.org/10.1007/978-1-4684-1716-6_25, 1962.

Wilks, D. S.: Statistical Methods in the Atmospheric Sciences, Academic Press, 1995.

Wylie, D., Jackson, D. L., Menzel, W. P., and Bates, J. J.: Trends in Global Cloud Cover in Two Decades of HIRS Observations, Journal of Climate, 18, 3021–3031, https://doi.org/10.1175/JCLI3461.1, 2005.